# LOW-FREQUENCY AND ROBUST IMAGE INFORMATION HIDING

## ABSTRACT

Existing image information hiding methods commonly lack robustness to varying degrees of distortion on container images. This paper proposes a low-frequency and robust image information hiding method, LRIIS, to overcome current challenges. To emphasize robustness, instead of hiding high-frequency subbands, we propose a novel wavelet contrastive loss to constrain so that most secret information is hidden in the low-frequency subbands. Compared to high-frequency subband hiding, low-frequency subband embedding achieves enhanced robustness. To alleviate the varying degrees of distortion influence, we build an unsupervised Attacked Image Enhancement Module (AEM) to generate the de-attacked image that is close to the corresponding container image. Notably, thanks to the pseudo-class label of AEM, the proposed method can recover the secret image from the attacked image without requiring a specific attack label. Experimental results demonstrate the superior performance of the proposed LRIIS model on the COCO and DIV2K datasets compared to existing state-of-the-art image information hiding methods.

## 1 INTRODUCTION

Image hiding Chanu et al. (2012) is a technique that conceals a secret image within a container image through the hiding process, allowing the secret image to be restored by the revealing process. During the hiding process, the secret and host images are used as input, and a container image is produced. The hiding process is imperceptible to human vision. Then, in the revealing process, the secret information will be extracted from the container image. The extracted image usually appears identical to the secret image. Image information hiding aims to maintain covertness while ensuring security and evading detection through steganalysis. However, existing image information hiding methods commonly lack robustness to distortion during the spread of the container image in the media Xu et al. (2022) and typically require a given label as a condition to alleviate the influence of distortion. Traditional image information hiding methods Mielikainen (2006) Cox et al.

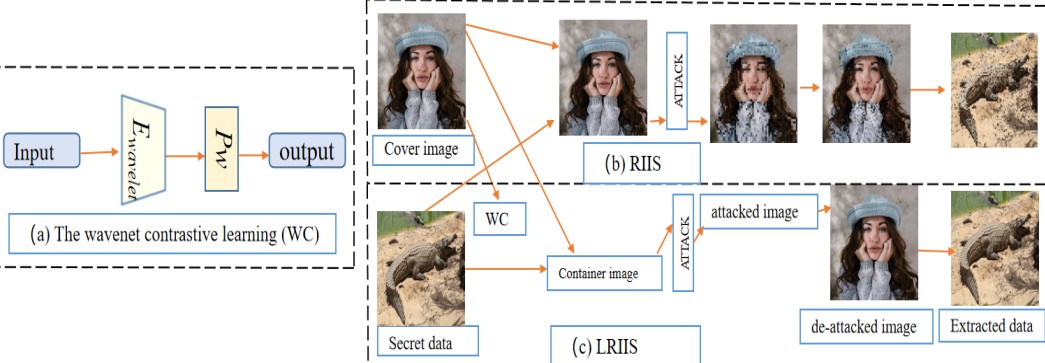

Figure 1: Previous high-frequency steganography, like RIIS $Xu\ et\ al.$ (2022), gains poorly extracted data and de-attacked image when the container is under different global masic attacks. On the contrary, our low-frequency method considers deep and diverse attacks, which demonstrates satisfactory robustness.

(1997) Lin et al. (2008) often suffer from poor quality and insufficient capacity to handle container images. In response, a new wave of methods based on Convolutional Neural Networks (CNN) has been attracting significant interest among researchers. In Baluja (2017) and Yang et al. (2019), they have attempted to minimize the impact of embedding on host images and introduced a cost-learning framework. These methods allow for the direct embedding of full-size secret images into the host images. In particular, they separate the hiding and revealing processes into two distinct steps and create two separate networks with independent parameters for each. Then HiNet Jing et al. (2021) was proposed to hide and reveal secret images with one single invertible neural network (INN). Its hiding and revealing processes share the same network parameters. However, HiNet is vulnerable to interference during the media spread of the container Xu et al. (2022). As a result, a Robust Invertible Image Information Hiding (RIIS) Xu et al. (2022) was proposed to alleviate the influence of distortion and emphasize robustness. However, RIIS is designed to work with a given attack label in certain distortions (e.g., different kinds of Gaussian, Poisson noise, or JPEG compression). It lacks robustness to varying degrees of attacks and typically requires a specific attack label as a condition to mitigate the distortion influence shown in Figure 1(b).

In Xiang et al. (2008), they proposed the intriguing idea of low-frequency subband embedding to achieve enhanced robustness. However, hiding in the low-frequency subbands is more visible and produces high-frequency artifacts Durall et al. (2020). For high-frequency artifacts, we speculate that there may be crossover subbands between the high-frequency subbands and the low-frequency subbands. The motivation is from Lin et al. (2024), where they introduced a K-times Dropout Neighboring Feature Transformer (KDNFT) to accept a set of neighboring features obtained by dropout as input to emphasize robustness. As a result, instead of hiding high-frequency subbands Xu et al. (2022) Jing et al. (2021) Yu et al. (2023) Chahine & Kim (2024) Lin et al. (2024), we propose a wavelet contrastive learning network(WC) to ensure that secret information is hidden in the low-frequency wavelet subbands for our INN frameworkshown in Figure 1(c). The INN framework and WC are trained simultaneously and share weight parameters. The wavelet contrastive loss is designed to force the high-frequency subbands away from the low-frequency subbands in WC and INN framework, shown in Figure 1(a). Then after the container image goes through a global masic attack, it will output the attacked image. To alleviate the varying degrees of distortion influence, we build an unsupervised Attacked Image Enhancement Module (AEM) to generate the de-attacked image that is close to its corresponding container image. In particular, thanks to the pseudo-class label of AEM, it is the first to recover the secret image from the attacked image without giving its corresponding attack label.

The main contributions of the proposed method are summarized as follows. First, instead of hiding high-frequency subbands, we propose a novel wavelet contrastive loss to force most secret information to be hidden in low-frequency subbands, which can alleviate the distortion influence. Secondly, to alleviate the varying degrees of distortion influence, we build an unsupervised Attacked Image Enhancement Module to generate the de-attacked image that is close to its corresponding container image. Finally, it is the first robust steganograph model to recover the secret image from the attacked image without giving its corresponding attack label.

## 2 RELATED WORKS

### 2.1 IMAGE INFORMATION HIDING

The concealment capabilities are typically restricted to lower than 0.4 bpp in most traditional image information hiding techniques. These methods typically involve the use of the spatial domain and transformation techniques of hiding information from the domain image. The common spatial domain hiding methods, including Mielikainen (2006), Yang et al. (2008), Luo et al. (2010), were initially developed to successfully embed additional data in noisy regions of a given host image. Puyang et al. embedded >0.4 bpp information in encrypted images via double MSB predictionPuyang et al. (2018). Puteaux et al. achieved payloads up to 2.45 bpp using recursive bit-plane prediction Puteaux & Puech (2020). As described in Pevny et al. (2010), Pevny et al. successfully hid the payload into the host image. To emphasize robustness, Lin et al. (2008) was introduced to hide secret data in the transform domain. However, the current methods used in these practices have been noted they have low quality and restricted embedding capacity. Researchers have recently begun to show considerable interest in CNN-based methods, as these methods can generate

a new container image instead of modifying the host image. In CNN-based works, Ma et al. Ma et al. (2018) introduced a new general information-hiding feature selection method, and Ye et al. Ye et al. (2017) proposed a new methodology to tackle information-hiding. Later, Wang et al. Wang et al. (2019) introduced a novel evaluation method for information hiding, which aims to provide a more reliable and accurate evaluation of information-hiding algorithms. In Boroumand et al. (2018), Boroumand et al. introduced a novel residual network, which can be applied both in the spatial domain and in JPEG information hiding. In Zhang et al. (2018), Zhang et al. introduced a novel method, leveraging the adversarial examples technique, to create improved hosts. Next, Hayes & Danezis (2017) and Tang et al. (2019) leverage the power of GANs model Goodfellow et al. (2014) to directly embed full-size images within the host images. Subsequently, Baluja (2017) developed a Hiding Network with auto-encoding capabilities, and Yang et al. (2019) designed a Hiding Network of the U-Net Ronneberger et al. (2015) type, which was accompanied by a discriminator. Often, hiding and revealing mechanisms are addressed as distinct processes, necessitating the implementation of two separate networks with different parameters to cater to them. Then ISN Lu et al. (2021) and HiNet Jing et al. (2021) were proposed to hide and reveal secret images with only one invertible neural network (INN). Its network parameters apply equally to both hiding and revealing processes. Nevertheless, these methods were susceptible to interference during the media's propagation of the container Yu et al. (2023) Chahine & Kim (2024) Lin et al. (2024) Chen et al. (2025). As a result, a Robust Invertible Image Information Hiding (RIIS) Xu et al. (2022) was proposed to alleviate the distortion influence and emphasize robustness. IN RoSteALS Bui et al. (2023), Bui et al. introduced a new idea for robust watermarking that does not require content specialization. The secret data is hidden in the latent code. However, current robust models were designed to work with a given attack label in certain distortions. They are vulnerable to varying degrees of global masked attacks on container images without a given attack label.

## 2.2 Contrastive Learning

Contrastive learning, as suggested by Hadsell et al. (2006), has recently seen a surge in interest in the field of image generation, especially as discussed by Dosovitskiy et al. (2014). Among the early adopters, Wu et al. Wu et al. (2018) utilized noise-contrastive estimation to encode instance class representations. In their recent study, He et al. introduced the Momentum Contrast (MoCo) He et al. (2020) approach that involves constructing a dynamic dictionary for unsupervised learning. Subsequently, Chen et al. Chen et al. (2020) provided a novel and simplistic framework, referred to as SimCLR, to contrast and improve upon existing representation methodologies. Previous approaches using supervised learning and semisupervised learning on the ImageNet dataset were outperformed by SimCLR, which successfully produced augmentation-invariant hiding for input images. Park et al. recently presented a new, multilayer, and patchwise contrastive learning model named CUT Park et al. (2020), which aims to enhance the capabilities of image-to-image translation using MoCo. In their paper entitled CoMoGAN Pizzati et al. (2021), Pizzati et al. introduced a continuous Generative Adversarial Network (GAN) that uses a functional manifold. In the study Han et al. (2021), DCUT employs two encoders to deal with unpaired data by using contrastive learning. In the work of Baek et al. Baek et al. (2021), a novel unsupervised image translation model was introduced. The key component of this model was Invariant Information Clustering (IIC) Ji et al. (2019), which was used to provide unsupervised domain classification. Patashnik et al. recently proposed a novel unsupervised translation network, BalaGAN Patashnik et al. (2021), which was specifically engineered to address the domain imbalance issue. In Cao et al. (2023), Cao et al. have proposed an innovative unsupervised translation model. This model is developed by applying conditional contrastive learning techniques to address the domain variations problem. To eliminate the influence of global masic attacks, we build a many-to-one class enhancement module that can unsupervisedly produce a pseudo-class label for each attacked image and use the pseudo-class label to remove global masic.

## 3 Proposed Method

The major target of LRIIS is to design a new low-frequency and robust invertible image information hiding framework for the distortion of global masic attacks. Let $\{h_i\}_{i=1}^n$, $\{s_i\}_{i=1}^n$, $\{hs_i\}_{i=1}^n$, $\{es_i\}_{i=1}^n$, $\{dhs_i\}_{i=1}^n$, $\{hs_i'\}_{i=1}^n$ denote the set of host, secret, container, extracted, attacked, and de-attacked images respectively. As shown in Figure 2(d), we construct an INN framework to hide the secret image in the container image and extract the secret image from the attacked image that is

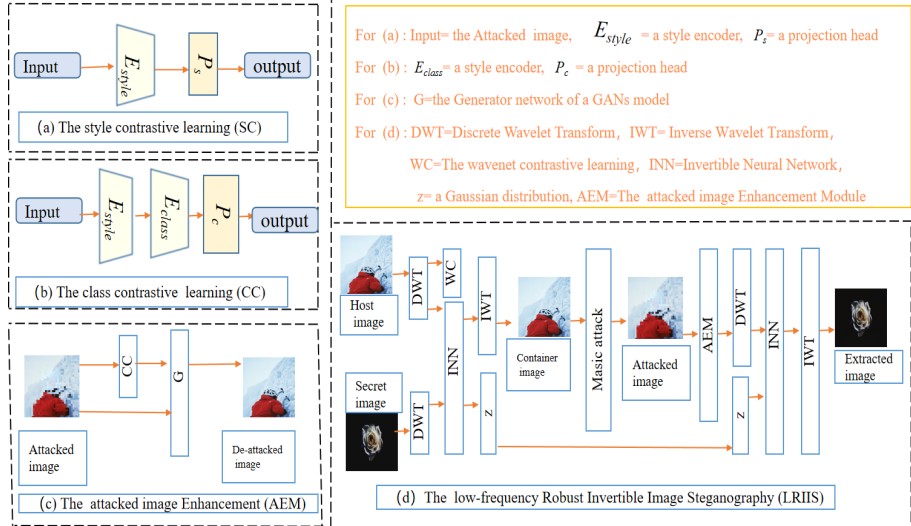

Figure 2: An overview of the proposed framework. (a) The style contrastive learning network. (b) The class contrastive learning network. (c) The Attacked Image Enhancement Module (AEM). (d) The low-frequency and Robust Invertible Image Information Hiding (LRIIS).

generated after the container image goes through a global masic attack. To alleviate the distortion influence of global masic attacks, we construct the AEM to generate the de-attacked image that is close to its corresponding container image shown in Figure 2(c). In practice, we have incorporated the AEM into the revealing process of the LRIIS.

### 3.0.1 ATTACKED IMAGE ENHANCEMENT MODULE

AEM is designed to generate the de-attacked image that is close to its corresponding container image.

**The process of global masic attacks.** Firstly, the container image $hs_i$ is partitioned into $K$ sub-blocks of size $j \times j$ each. For each sub-block, the RGB value of the top-left pixel is used as the representative value for the entire block. Afterward, all the sub-blocks are cyclically processed to produce the targeted image $dhs_i$. When $K$ changes between $m$ and $n$, given the container image $hs_i$, it will produce a set of attacked images $\{dhs_{ib}\}_{b=m}^{n}$.

**The process of Attacked Image Enhancement.** Based on the global masic attack process, given the container image $hs_i$, $n - m$ classes of global masic attacks must be classified and alleviated. Given the attacked image $dhs_{ib}$, if the attack label $K$ is not given, eliminating the distortion of global masic attacks is a challenging task for RIIS Xu et al. (2022). AEM breaks this challenge down into many-to-one class enhancement problems and can unsupervisedly produce a pseudo-class label for each attacked image $dhs_{ib}$, and uses the pseudo-class label to eliminate distortion. During the AEM process, two distinct stages are involved. During the first stage, an unsupervised semantic clustering network is developed to create a pseudo-class label for each attacked image $dhs_{ib}$. We implement two unsupervised encoders in the semantic clustering network: a style encoder $E_{style}$ followed by a classified encoder $E_{class}$. In the process of AEM, the SimCLR model by Chen et al. is utilized as a pretext task. As depicted in Figure 2(a), the style contrastive learning network is a system composed of two main components: the style encoder $E_{style}$ and a projection head $P_s$. Then, AEM utilizes the attacked image $dhs_{ib}$ and its corresponding container image $hs_i$ (positive samples) to compute the style contrastive loss. In terms of a given batch size $B$, the remaining $2(B-1)$ attacked images (negative samples) form a set $\mathcal{F}(i)$ together. The style contrastive loss $\mathcal{L}_s$ can be written as:

$$\mathcal{L}_s = -\log \frac{\exp\left(\text{sim}\left(q_i, q_i'\right)/\tau\right)}{\sum_{j \in \mathcal{J}(i)} \exp\left(\text{sim}\left(q_i, q_j\right)/\tau\right)}, \quad \mathcal{J}(i) \equiv \{i', n \mid n \in \mathcal{F}(i)\} \tag{1}$$

where $\{q_i\} = \{P_s\left(E_{style}(dhs_{ib})\right)\}$ and $\{q_i'\} = \{P_s\left(E_{style}(hs_i)\right)\}$, $\text{sim}(,)$ is used to calculate cosine similarity, $\tau$ means temperature parameter and we set $\tau = 0.07$. The AEM then classifies the style representations of all attacked images $\{dhs_{ib}\}_{b=m}^{n}$ via a classified encoder. The attacked

images in each cluster share a similar style class. The class contrastive learning network is a system composed of two main components: a classified encoder, $E_{class}$, and a projection head, $P_c$ (as depicted in Figure 2(b)). AEM constrains each attacked image $dhs_{ib}$ and its nearest neighbors $N_i = \{dhs_{io}\}_{o=H}^{L}$ to be best categorized as a unique class rather than be split into the remaining $n - m - 1$ classes. Given the batch size $B$, the total number of $2(B-1)$ attacked images pass $E_{class}$ and $E_{style}$, and it will produce the pseudo labels $\mathcal{C}(i)$. The class contrastive loss is given by:

$$\mathcal{L}_{c} = -\log \frac{\exp\left(\mathrm{sim}\left(p_i, p_i^N\right)/\tau\right)}{\sum_{j \in \mathcal{J}(i)} \exp\left(\mathrm{sim}\left(p_i, p_j^N\right)/\tau\right)}, \quad \mathcal{J}(i) \equiv \{i, n \mid n \in \mathcal{C}(i)\} \tag{2}$$

where $\{p_i\} = \{P_c(E_{class}(E_{style}(dhs_{ib})))\}$ and $\{(p_i^N)\} = \{P_c(E_{class}(E_{style}(N_i)))\}$.

In the second stage of AEM, a many-to-one class enhancement network is trained by using the pseudo-class label from the first stage to eliminate the distortion for each attacked image $dhs_{ib}$. Here, AEM assumes that the attacked images $\{dhs_{ib}\}_{b=m}^{n}$ contain $n - m$ classes, and the container image $\{hs_i\}$ contains 1 class. In a generator network $G$, given the attacked image $dhs_{ib}$, and the container image $hs_i$, the de-attacked image is given by:

$$hs_i{}' = G(E_{class}(E_{style}(dhs_{ib})), hs_i) \tag{3}$$

In this way, given each attacked image $dhs_{ib}$, AEM will output the de-attacked image $hs_i'$ that is close to its corresponding container image $hs_i$ as shown in Figure 2(c).

### 3.0.2 LOW-FREQUENCY AND ROBUST IMAGE INFORMATION HIDING FRAMEWORK

Here, we design a robust and low-frequency invertible image information hiding framework to hide and reveal secret images. We introduce a wavelet contrastive learning network(WC) to ensure that secret information is hidden in the low-frequency wavelet subbands for our INN framework. The INN framework and WC are trained simultaneously and share weight parameters. The INN-based framework consists of a hiding and revealing process. In the Hiding process, the host image $h_i$ and the secret image $s_i$ are first converted to the frequency subbands using the discrete wavelet transform (DWT), and then it passed through $M$ hiding blocks. Given the inputs for the $i$th hiding block in the forward process, $h_i$ and $s_i$, the outputs, $h_{i+1}$ and $s_{i+1}$, are determined as follows:

$$\begin{aligned} \boldsymbol{h}_{i+1} &= \boldsymbol{h}_i + \phi\left(\boldsymbol{s}_i\right) \\ \boldsymbol{s}_{i+1} &= \boldsymbol{s}_i \odot \exp\left(\alpha\left(\boldsymbol{\rho}\left(\boldsymbol{h}_{i+1}\right)\right)\right) + \boldsymbol{\eta}\left(\boldsymbol{h}_{i+1}\right) \end{aligned} \tag{4}$$

where $\alpha$ is used to prevent values from exceeding a certain range by a sigmoid function, and $\odot$ means the operation of the dot product. Here, $\rho(\cdot), \phi(\cdot)$ and $\eta(\cdot)$ are arbitrary functions. After the final hiding block, it produces $\boldsymbol{h}_{M+1}$ and $\boldsymbol{s}_{M+1}$. Repeat this process, and it will output two images. An image is further converted to the spatial domain by the inverse wavelet transform (IWT), resulting in the container image $hs_i$. The other image $\hat{z}$ follows a Gaussian distribution, and we can use it in the revealing process. In practice, the container image $hs_i$ will be attacked by different global masic attacks and becomes an attacked image $dhs_ib$. In the revealing process, after $dhs_ib$ is enhanced by AEM, it will output the de-attacked image $hs_i'$. Given $\boldsymbol{hs}_{i+1}'$ and $\boldsymbol{z}^{i+1}$ as input for the $i$-th revealing block and $\boldsymbol{hs}_i'$ and $\boldsymbol{z}^i$ as output, this process can be represented as follows:

$$\begin{aligned} \boldsymbol{z}^i &= \left(\boldsymbol{z}^{i+1} - \boldsymbol{\eta}\left(\boldsymbol{hs}_{i+1}'\right)\right) \odot \exp\left(-\alpha\left(\boldsymbol{\rho}\left(\boldsymbol{hs}_{i+1}'\right)\right)\right) \\ \boldsymbol{hs}_i' &= \boldsymbol{hs}_{i+1}' - \phi\left(\boldsymbol{z}^i\right). \end{aligned} \tag{5}$$

Once the last revealing block is processed, the resulting output, $\boldsymbol{hs}_i'$, is fed into an IWT block to produce the extracted image $\boldsymbol{es}_i$.

### 3.0.3 LOSS FUNCTION

The loss function in this paper is composed of three integral parts: the AEM loss is to mitigate the impact of global masic attacks, the HR loss is to ensure the hiding and revealing performance, and a unique wavelet contrastive loss is designed to ensure that secret information is specifically concealed within the lower-frequency wavelet subbands.

The AEM loss aims to ensure that the output de-attacked image $hs_i'$ appears identical to the container image $hs_i$. Toward this goal, the AEM loss $\mathcal{L}_{\text{AEM}}$ is defined as follows:

$$\mathcal{L}_{\text{AEM}} = \left\| hs_i' - hs_i \right\|_2 \tag{6}$$

It is required that the container image $hs_i$ and extracted image $es_i$ appear identical to the input host image $h_i$ and the secret image $s_i$. We also constrain $\mathbf{z}$ to a Gaussian distribution and depict the distribution distance by the cross-entropy (CE). Toward this goal, the HR loss $\mathcal{L}_{\text{HR}}$ is defined as follows:

$$\mathcal{L}_{\text{HR}} = \|hs_i - h_i\|_2 + \|es_i - s_i\|_2 + \ell_{\mathcal{CE}}(p(\mathbf{z}), \mathcal{N}(\mathbf{0}, \mathbf{I})) \tag{7}$$

Furthermore, a novel wavelet contrastive loss is proposed to guarantee that secret information is subtly concealed within the low-frequency wavelet subbands instead of the high-frequency subbands Xu et al. (2022) Jing et al. (2021). We speculate to suggest the presence of crossover subbands between the high-frequency subbands and the low-frequency subbands. The wavelet contrastive loss is designed to force the high-frequency subbands away from the low-frequency subbands, and it narrows the crossover subbands as much as possible. Suppose that $\mathcal{H}(\cdot)_{LL}$, $\mathcal{H}(\cdot)_{HH}$ indicate the operation of extracting low-frequency subbands, high-frequency subbands after wavelet decomposition. The proposed wavelet contrastive learning network chooses SimCLRChen et al. (2020) as a pretext task and consists of an encoder $E_{wavelet}$ and a projection head $P_w$. The high-frequency information of the container image, the host image can be written as $\mathcal{H}(hs_i)_{HH}$, $\mathcal{H}(h_i)_{HH}$ (positive samples). Given the batch size $B$, the low-frequency information of the host image $\mathcal{H}(h_i)_{LL}$ ($2(B-1)$ negative samples) form $\mathcal{F}(i)$. Finally, the contrastive loss of wavelets is designed to constrain that $\mathcal{H}(hs_i)_{HH}$ is close to $\mathcal{H}(h_i)_{HH}$ but far from $\mathcal{H}(h_i)_{LL}$. The wavelet contrastive loss $L_{\text{CT}}$ can be defined as:

$$\mathcal{L}_{\text{CT}} = -\log \frac{\exp\left(\text{sim}\left(w_i, w_i'\right)/\tau\right)}{\sum_{j \in \mathcal{J}(i)} \exp\left(\text{sim}\left(w_i, w_j\right)/\tau\right)}, \quad \mathcal{J}(i) \equiv \{i', n \mid n \in \mathcal{F}(i)\} \tag{8}$$

where $\{w_i\} = \{P_w\left(E_{wavelet}(\mathcal{H}(hs_i)_{LL})\right)\}$ and $\{w_i'\} = \{P_w\left(E_{wavelet}(\mathcal{H}(h_i)_{LL})\right)\}$, $\text{sim}(,)$ is used to calculate cosine similarity, $\tau$ means temperature parameter and we set $\tau = 0.07$.

In summary, the total loss function considers the following three components: the AEM loss, the HR loss, and the CT loss:

$$\mathcal{L}_{\text{total}} = \lambda_1 \mathcal{L}_{\text{AEM}} + \lambda_2 \mathcal{L}_{\text{HR}} + \lambda_3 \mathcal{L}_{\text{CT}} \tag{9}$$

Wherein, the parameter values of $\lambda_1$ and $\lambda_2$ are set to 6 in the training process, while $\lambda_3$ is set to 10. To guarantee that secret information is embedded within the low-frequency wavelet subbands, this paper selects the parameter value 10 for the wavelet contrastive learning loss. Since there is a higher probability that the AEM loss and the HR loss will converge, this paper sets a specific parameter value of 6 for them.

## 4 EXPERIMENTS

To explore the ability of LRIIS, we train and test the method on two datasets and compare it with four state-of-the-art (SOTA) methods.

### 4.1 DATASETS AND SETTINGS

This paper uses a total of 2 datasets, DIV2K Agustsson & Timofte (2017) and COCOChen et al. (2015), to train and test the model. DIV2K is an extensive collection of RGB images with an abundance of diverse content. Additionally, COCOChen et al. (2015) serves as a robust dataset that is particularly suitable for tasks such as image translation and object segmentation. For each dataset, we have a total of 800 images that are utilized for training and an additional 500 images that are used for evaluation. The used images are uniformly resized to a resolution of $1024 \times 1024$ pixels. In the training process, we input 400 host/secret image pairs and use the Adam optimizer with $\beta_1 = 0.9$ and $\beta_2 = 0.99$. The batch size is set to 4 for training, and the learning rate is 0.0001. The number of hiding and revealing blocks $M$ is set to 16. To ensure that AEM is sufficient to generate the correct pseudo-class label for each attacked image, we set the attack labels $m = 200, n = 600$

for $\{dhs_{ib}\}_{b=m}^n$. The process requires approximately 20 hours for training 80 epochs on a single GTX3090ti GPU. Specifically, it takes about 12 hours to pre-train AEM and about 8 hours to train the INN framework.

## 4.2 EXPERIMENTAL RESULTS

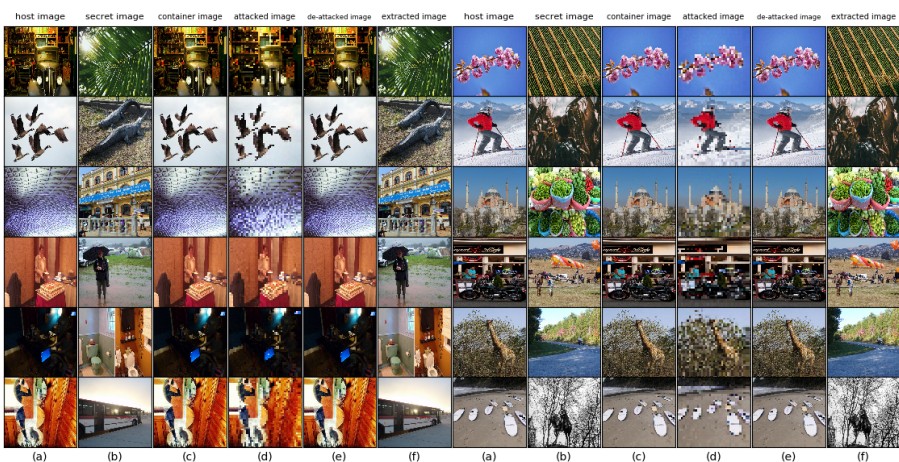

Figure 3: The image hiding results of LRIIS. (a) The host image. (b) The secret image. (c) The container image. (d) The attacked image. (e) The de-attacked image. (f) The extracted image.

Figure 3 shows the corresponding information-hiding results for the DIV2K and COCO datasets, presented in the first three and last three rows, respectively. It is evident from the illustration that the proposed LRIIS model can create visually stunning and accurate container images that reflect their true appearance from the host images. This capability serves as a testament to its effectiveness in hiding secret images within the lower-frequency wavelet subbands. Furthermore, the extracted secret images from the attacked images depicted in Figure 2 are strikingly similar to the original secret photos. It proves that AEM has the power to alleviate the distortion influence of global masic attacks with different attack labels. Importantly, it should be noted that the de-attacked images also display a high degree of similarity to the container images.

## 4.3 BASELINES

To ensure the validity of LRIIS, we compared it against four current SOTA image-hiding methods, namely HiNet Jing et al. (2021), HIIDM (Hiding images in diffusion models) Chen et al. (2025), RIIS Xu et al. (2022), and CrossNET Yu et al. (2023). For a fair comparison, we retrained all of the models by using the same training dataset as ours.

## 4.4 EVALUATION METRICS

We conduct a quantitative analysis and compare our proposed method to these established baselines using four evaluation metrics. These metrics include PSNR, SSIM, RMSE, and MAE. Here, PSNR denotes the Peak Signal-to-Noise Ratio Hore & Ziou (2010), while SSIM represents the Structural Similarity Index Wang et al.. As we know, increasing PSNR and SSIM values means enhanced image quality. Here, RMSE stands for Root Mean Square Error, while MAE denotes Mean Absolute Error. In the same way, decreasing MAE and RMSE values means enhanced image quality.

## 4.5 COMPARISON WITH SOTA METHODS

In this section, we will evaluate the proposed method against existing SOTA methods. The comparison is carried out both qualitatively and quantitatively. Note that HIIDM and HiNet are more vulnerable to interference during container media spread Xu et al. (2022), so we add the AEM in their revealing process. We also gave the attack labels for RIIS and CrossNET to alleviate the distortion influence. Figure 3 shows the comparison results of information-hiding experimental tests on two datasets, which were randomly selected.

Figure 4: The experiment comparison results. (a) host images. (b) $\text{HiNet}_C$ indicates the container of HiNet. (c) $\text{HIIDM}_C$ indicates the container of HIIDM. (d) $\text{RIIS}_C$ indicates the container of RIIS. (e) $\text{CrossNET}_C$ indicates the container of the CrossNET. (f) $\text{ours}_C$ indicates the container of LRIIS. (g) secret images. (h) $\text{HiNet}_S$ indicates the extracted image of HiNet. (i) $\text{HIIDM}_S$ indicates the extracted images of HIIDM. (j) $\text{RIIS}_S$ indicates the extracted images of RIIS. (k) $\text{CrossNET}_S$ indicates the extracted images of CrossNET. (l) $\text{ours}_S$ indicates the extracted images of LRIIS.

**Qualitative results.** In Figure 4, the corresponding image information-hiding comparison results for both the DIV2K and COCO datasets are illustrated. The first three rows display the results for the DIV2K dataset, while the remaining three rows correspond to the COCO dataset. In this comparison, the container images of the proposed method ($\text{ours}_C$) demonstrate superior performance compared to HIIDM ($\text{HIIDM}_C$), HiNet ($\text{HiNet}_C$), RIIS ($\text{RIIS}_C$), and the CrossNET method ($\text{CrossNET}_C$) on two datasets. One clear issue is that there is a significant visual disparity between the existing models' container images and their corresponding host images. The findings presented in Figure 3 further demonstrate that the proposed method ($\text{ours}_S$) delivers superior performance compared to the HIIDM method ($\text{HIIDM}_S$), the HiNet method ($\text{HiNet}_S$), the RIIS method ($\text{RIIS}_S$), and the CrossNET method ($\text{CrossNET}_S$) on both datasets. It proves that existing image methods are vulnerable to distortion of global masic attacks with different attack labels. One main reason is that it is more stable when the secret information is hidden in the low-frequency wavelet subbands, and it enhances the overall security of the information hiding process. Moreover, current models are designed for the distortion of Gaussian noise, Poisson noise, and JPEG compression with one attack label, but not for different attack labels.

**Quantitative evaluations.** This paper not only presents visually impressive results but also incorporates a quantitative evaluation using four metrics across two datasets.

Table 1: Quantitative evaluations in terms of two datasets.

| * Methods | host/container image pair | | | | | | | |
|---|---|---|---|---|---|---|---|---|
| | DIV2K dataset | | | | COCO dataset | | | |
| | PSNR ↑ | SSIM ↑ | MAE ↓ | RMSE ↓ | PSNR ↑ | SSIM ↑ | MAE ↓ | RMSE ↓ |
| HiNet | 30.69 | 0.9013 | 8.91 | 8.01 | 28.69 | 0.8843 | 9.21 | 9.75 |
| HIIDM | 32.11 | 0.9191 | 7.63 | 7.72 | 30.57 | 0.8996 | 8.92 | 8.01 |
| RIIS | 35.43 | 0.9321 | 6.42 | 6.50 | 33.08 | 0.9105 | 7.51 | 7.33 |
| CrossNET | 38.19 | 0.9597 | 4.12 | 4.31 | 36.49 | 0.9417 | 5.03 | 5.6 |
| ours | **41.99** | **0.9821** | **2.53** | **1.94** | **39.52** | **0.9711** | **2.92** | **3.26** |
| * Methods | secret/extracted image pair | | | | | | | |
| | DIV2K dataset | | | | COCO dataset | | | |
| | PSNR ↑ | SSIM ↑ | MAE ↓ | RMSE ↓ | PSNR ↑ | SSIM ↑ | MAE ↓ | RMSE ↓ |
| HiNet | 32.56 | 0.9207 | 7.50 | 7.56 | 30.01 | 0.8975 | 8.99 | 8.52 |
| HIIDM | 34.11 | 0.9435 | 6.17 | 5.69 | 32.46 | 0.9245 | 7.76 | 8.03 |
| RIIS | 36.48 | 0.9501 | 4.57 | 4.63 | 34.31 | 0.9429 | 6.58 | 6.89 |
| CrossNET | 39.58 | 0.9617 | 3.68 | 3.11 | 37.39 | 0.9596 | 4.28 | 4.32 |
| ours | **42.82** | **0.9876** | **1.95** | **1.72** | **40.51** | **0.9727** | **3.32** | **2.47** |

The proposed method yields superior performance, as compared to the SOTA deep learning-based methods, in terms of the average PSNR, SSIM, MAE, and RMSE values, as shown in Table 1. For the host/container image pair, we reach (41.99/0.9821/2.53/1.94) in the DIV2K dataset and (39.52/0.9711/2.92/3.26) in the COCO dataset. For the secret/extracted image pair, we reach (42.82/0.9876/1.95/1.72) in the DIV2K dataset and (40.51/0.9727/3.32/2.47) in the COCO dataset. It means that existing image information hiding methods commonly lack robustness to distortion of global masic attacks with different attack labels. The reason for the best score is that LRISS successfully narrows the crossover subbands between the high-frequency subbands and the low-frequency subbands.

## 4.6 ABLATION STUDY

**Effectiveness of the wavelet contrastive loss**. The wavelet contrastive loss is to ensure that most secret information is hidden in the low-frequency wavelet subbands. The proposed model can be written as $LRIIS_1$ when the wavelet contrastive loss is removed. Some test results for $LRIIS_1$ and LRIIS are shown in Figure 5:

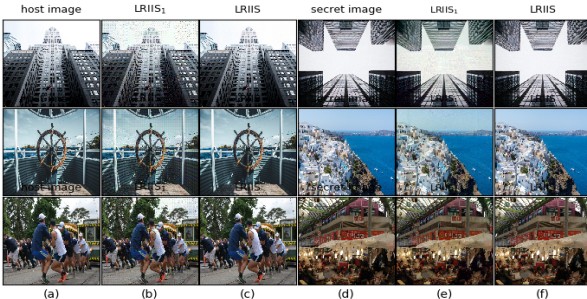

Figure 5: Ablation study. (a) host images. (b) The container images of $LRIIS_1$. (c) The container images of LRIIS. (d) secret images. (e) The extracted images of $LRIIS_1$. (f) The extracted images of LRIIS.

As shown in Figure 5, when the wavelet contrastive loss has been removed, the container images of $LRIIS_1$ are significantly different from the input host images, and the extracted images of $LRIIS_1$ are not similar to the input secret images. Obviously, without the wavelet-contrasting loss, hiding in the low-frequency subbands is more visible and produces high-frequency artifacts. One main reason is that the wavelet contrastive loss successfully forces the high-frequency subbands away from the low-frequency subbands, and it narrows the crossover subbands as much as possible.

For a similar result of the DIV2K dataset and the COCO dataset, only the ablation study results of the DIV2K dataset are shown in Table 2. As demonstrated, the wavelet contrastive loss significantly improves the security of our method. For the host/container image pair and the secret/extracted image pair, all the scores have been greatly improved.

Table 2: Ablation study in the DIV2K dataset.

| * Methods | host/container image pair | | | | secret/extracted image pair | | | |
|---|---|---|---|---|---|---|---|---|
| | DIV2K dataset | | | | DIV2K dataset | | | |
| | PSNR ↑ | SSIM ↑ | MAE ↓ | RMSE ↓ | PSNR ↑ | SSIM ↑ | MAE ↓ | RMSE ↓ |
| $LRIIS_1$ | 32.36 | 0.9195 | 7.58 | 7.63 | 34.22 | 0.9451 | 6.14 | 5.65 |
| LRIIS | **41.99** | **0.9821** | **2.53** | **1.94** | **42.82** | **0.9876** | **1.95** | **1.72** |

## 5 CONCLUSION

This paper presents a new low-frequency and robust image information-hiding method designed to tackle the difficulties of image information hiding. We introduce a novel wavelet contrastive loss function, thus targeting to confine the most secret information to the low-frequency subbands, which significantly enhances the robustness of the hiding process. To eliminate the distortion of global masic attacks, we build an unsupervised Attacked Image Enhancement Module to successfully generate

the de-attacked image that is close to its corresponding container image. Notably, it is the first time to recover the secret image from the attacked image without giving its corresponding attack labels. This proposed method has excelled the existing state-of-the-art methods, showing impressive results for image information hiding on two datasets.

**Ethics Statement and Reproducibility Statement**.

## ETHICS STATEMENT

This paper presents a low-frequency and robust image information hiding method (LRIIS) for improving the robustness of image information hiding against global masic attacks. Throughout the research and development process of this work, we have strictly adhered to ethical principles and legal regulations to ensure that the proposed method is used in a responsible and secure manner.

First, regarding data usage, the experiments in this paper are conducted on two publicly available datasets: DIV2K and COCO. Both datasets are widely recognized in the computer vision community and have clear usage licenses that permit academic research. We have not modified or repurposed the data beyond the scope allowed by the dataset licenses, nor have we collected or used any private, sensitive, or unlicensed image data. This ensures that the data used in our research does not infringe on the privacy rights or intellectual property rights of any individual or organization.

Second, concerning the potential applications of the proposed method, image information hiding technology has dual-use properties. It can be applied to legitimate scenarios such as copyright protection of digital media (e.g., embedding invisible watermarks to trace the source of images), secure transmission of confidential information in legal business or government contexts, and preservation of sensitive data in medical or military fields. However, it may also be misused for malicious purposes, such as hiding illegal content (e.g., obscene, violent, or terrorist-related images) to evade detection, or conducting covert communication for criminal activities. We explicitly oppose any malicious use of our proposed method. We advocate that users of this technology must comply with relevant national laws and ethical norms, and we will not provide technical support or guidance for any illegal or unethical applications.

Third, in the research process, we have maintained scientific integrity and transparency. All experimental results presented in this paper are based on repeated and verifiable experiments, and we have not fabricated, falsified, or exaggerated any data or results. We have also properly cited the work of other researchers in the references section, respecting the intellectual property rights of previous studies and avoiding plagiarism or improper use of others' achievements.

Finally, we recognize the social responsibility of technological innovation. As researchers in the field of information security and computer vision, we will continue to pay attention to the ethical implications and social impacts of image information hiding technology. We are willing to participate in discussions and collaborations with relevant institutions, policymakers, and the public to promote the formulation of ethical guidelines and regulatory frameworks for this technology, ensuring that it contributes positively to the development of society while minimizing potential risks and harms.

## REPRODUCIBILITY STATEMENT

To ensure the reproducibility of the experimental results and the proposed LRIIS method, we provide detailed information on data, code, experimental configurations, and implementation details as follows, in compliance with the reproducibility requirements of ICLR 2026.

### 1. DATASETS AND PREPROCESSING

- **Datasets Source**: All experiments use two publicly available datasets:
  - DIV2K (Agustsson & Timofte, 2017): Available at `https://data.vision.ee.ethz.ch/cvl/DIV2K/` (accessed in accordance with the dataset's open license).
  - COCO (Chen et al., 2015): Available at `https://cocodataset.org/` (used under the dataset's academic research license).
- **Data Splitting**: For each dataset, we use 800 images for training and 500 images for evaluation. The splitting is based on the default training/validation partitions of the datasets, with no additional random selection biases.
- **Preprocessing Steps**: All images are uniformly resized to a resolution of 1024×1024 pixels using bicubic interpolation. No other data augmentation operations (e.g., rotation, flipping,

or color jitter) are applied during training or evaluation, to ensure consistency in experimental conditions.

## 2. CODE AVAILABILITY

- **Code Repository**: We will make the complete source code of the LRIIS method publicly available on GitHub (or a similar open-source platform) upon the acceptance of this paper. The repository will include:

  - Implementation of the Invertible Neural Network (INN) for hiding and revealing processes (Section 3.0.2).
  - Code for the Attacked Image Enhancement Module (AEM), including the unsupervised semantic clustering network and the many-to-one class enhancement network (Section 3.0.1).
  - Implementation of the wavelet contrastive loss, AEM loss, and HR loss (Section 3.0.3).
  - Scripts for data loading, model training, evaluation, and result visualization.

- **Dependency Details**: The code is implemented using Python 3.8+ and PyTorch 1.10+. The required dependencies are listed in a `requirements.txt` file, including:

  - PyTorch 1.10.0, TorchVision 0.11.1 (for model building and training).
  - NumPy 1.21.2 (for numerical computations).
  - OpenCV-Python 4.5.3 (for image reading, resizing, and preprocessing).
  - Scikit-image 0.18.3 (for calculating evaluation metrics: PSNR, SSIM, MAE, RMSE).
  - Matplotlib 3.4.3 (for result visualization).
  - PyWavelets 1.1.1 (for discrete wavelet transform (DWT) and inverse wavelet transform (IWT)).

## 3. EXPERIMENTAL CONFIGURATIONS

- **Hardware Setup**: All experiments are conducted on a single NVIDIA GTX 3090 Ti GPU (24GB VRAM) with an Intel Core i9-12900K CPU and 64GB DDR4 RAM. No distributed training or multi-GPU acceleration is used, to simplify reproducibility.

- **Training Parameters**:

  - **Optimizer**: Adam optimizer with $\beta_1 = 0.9$ and $\beta_2 = 0.99$.
  - **Learning Rate**: Fixed at 0.0001 for the entire training process (no learning rate scheduling is applied).
  - **Batch Size**: 4 (for both training the AEM and the INN framework).
  - **Number of Epochs**: Total training takes 80 epochs, including 12 hours of pre-training for AEM and 8 hours of training for the INN framework.
  - **Hyperparameters**:
    * Number of hiding and revealing blocks ($M$): 16 (Section 3.0.2).
    * Attack labels range for global masic attacks: $m = 200$, $n = 600$ (Section 4.1), generating attacked images $\{dhs_{ib}\}_{b=m}^{n}$.
    * Temperature parameter ($\tau$) for contrastive losses (style contrastive loss, class contrastive loss, and wavelet contrastive loss): 0.07 (Sections 3.0.1 and 3.0.3).
    * Loss weights: $\lambda_1 = 6$ (AEM loss), $\lambda_2 = 6$ (HR loss), $\lambda_3 = 10$ (wavelet contrastive loss) (Section 3.0.3).

- **Evaluation Metrics**: All evaluations use four metrics, computed using standard implementations:

  - PSNR (Peak Signal-to-Noise Ratio): Calculated using `skimage.metrics.peak_signal_noise_ratio` (data range set to 255).
  - SSIM (Structural Similarity Index): Calculated using `skimage.metrics.structural_similarity` (data range set to 255, win_size=11).

- MAE (Mean Absolute Error): Computed as the average of absolute differences between pixel values of two images.
- RMSE (Root Mean Square Error): Computed as the square root of the average of squared differences between pixel values of two images.

## 4. BASELINE METHODS AND COMPARISON

- **Baseline Implementations**: To ensure a fair comparison, we re-implement four state-of-the-art (SOTA) methods using the same data and experimental setup as LRIIS:
    - HiNet (Jing et al., 2021): Implemented based on the original paper's INN architecture, with no AEM (unless specified otherwise in the ablation study).
    - HIIDM (Chen et al., 2025): Re-implemented using the diffusion model-based hiding framework described in the original paper.
    - RIIS (Xu et al., 2022): Implemented with attack label input (as required by the original method) to alleviate distortion.
    - CrossNET (Yu et al., 2023): Re-implemented following the diffusion model-controlled steganography approach in the original paper.
- **Comparison Setup**: For methods vulnerable to container image distortion (HiNet and HIIDM), we add the AEM module to their revealing process (as described in Section 4.5) to maintain consistent experimental conditions. All baselines are trained for the same number of epochs (80) with the same batch size (4) and optimizer settings as LRIIS.

## 5. ADDITIONAL IMPLEMENTATION DETAILS

- **Global Masic Attack Simulation**: The global masic attack is implemented by partitioning the container image ($hs_i$) into $K$ subblocks of size $j \times j$ ($j$ is set to 8, a default value that balances attack severity and computational efficiency). For each subblock, the RGB value of the top-left pixel is used as the representative value for the entire block, and cyclic processing of subblocks generates the attacked image ($dhs_{ib}$) (Section 3.0.1).
- **Wavelet Transform**: Discrete Wavelet Transform (DWT) and Inverse Wavelet Transform (IWT) use the "haar" wavelet basis (a common choice for image processing) from the PyWavelets library. The wavelet decomposition level is set to 1, to split images into low-frequency (LL) and high-frequency (HH, HL, LH) subbands.
- **Pseudo-Class Label Generation in AEM**: The unsupervised semantic clustering network uses SimCLR (Chen et al., 2020) as the pretext task. The style encoder ($E_{\text{style}}$) and class encoder ($E_{\text{class}}$) are both 4-layer convolutional neural networks (CNNs) with ReLU activation functions. The projection heads ($P_s$ for style contrastive loss, $P_c$ for class contrastive loss) are 2-layer fully connected networks (Section 3.0.1).

By following the above information, other researchers should be able to fully reproduce the experimental results of the LRIIS method, including quantitative metrics (Table 1, Table 2) and qualitative comparisons (Figures 3, 4, 5). For any ambiguities or technical issues during reproduction, we will provide timely support through the public code repository's issue section.

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

## A  THE CALCULATION FORMULA FOR BITS PER PIXEL (BPP)

Our work falls under the category of image-to-image steganography, where a full-size secret image is embedded into a host image. The payload is fixed as:

$$\text{bpp} = \frac{\text{Total bits of secret image}}{\text{Total pixels of container image}} = \frac{3 \times H \times W \times 8}{H \times W} = 24 \text{ bits per RGB pixel triplet} = 3 \text{ bpp}.$$

This differs fundamentally from traditional low-capacity steganography ($<$0.4 bpp) in that we aim for high capacity with robust reconstruction rather than subtle embedding.

To explore whether low-frequency embedding limits capacity, we design a variant LRIIS-HighFreq (forcing secrets into high frequencies) and compare its robustness at the same bpp:

Table 3: Comparison of different methods in terms of bpp and PSNR values for Mosaic and JPEG compression.

| Method | bpp | Mosaic ($K = 400$) PSNR | JPEG (QF=50) PSNR |
|---|---|---|---|
| LRIIS-HighFreq | 3 | 32.1 | 28.4 |
| LRIIS (Ours) | 3 | 40.5 | 37.6 |

Conclusion: Low-frequency embedding does not sacrifice capacity but enhances robustness.

## B  THE STATISTICAL SIGNIFICANCE TESTING

We performed a paired t-test on the COCO test set (500 images) comparing LRIIS with the strongest baseline CrossNET, testing the null hypothesis $H_0$ that there is no difference in average PSNR between the two methods.

Let $\{x_i\}_{i=1}^{500}$ denote the PSNR values of LRIIS, and $\{y_i\}_{i=1}^{500}$ those of CrossNET. Define the differences $d_i = x_i - y_i$. The test statistic is:

$$t = \frac{\bar{d}}{s_d/\sqrt{n}}, \quad \text{where } \bar{d} = \frac{1}{n}\sum_{i=1}^{n} d_i, \ s_d = \sqrt{\frac{1}{n-1}\sum_{i=1}^{n}(d_i - \bar{d})^2}.$$

Experimental results: $\bar{d}_{\text{PSNR}} = 3.12$ dB, $s_d = 1.87$, $t = 37.24$, degrees of freedom $df = 499$, critical value $t_{0.005,499} \approx 2.586$.

Since $|t| \gg 2.586$, we reject $H_0$ at $p < 10^{-10}$, confirming the improvement is highly significant.

A table can represent the average Significance Testing results as:

Table 4: Quantitative Evaluation on the COCO Dataset (Secret/Extracted Image Pairs, Mean $\pm$ Std Dev).

| Method | PSNR $\uparrow$ | SSIM $\uparrow$ | MAE $\downarrow$ | RMSE $\downarrow$ |
|---|---|---|---|---|
| CrossNET | $37.39 \pm 1.92$ | $0.9596 \pm 0.008$ | $4.28 \pm 0.31$ | $4.32 \pm 0.29$ |
| Our Method (LRIIS) | $\mathbf{40.51 \pm 1.03}$ | $\mathbf{0.9727 \pm 0.004}$ | $\mathbf{3.32 \pm 0.18}$ | $\mathbf{2.47 \pm 0.15}$ |

Our method outperforms the current state-of-the-art (SOTA) algorithm CrossNET, achieving the best results.

## C  LIMITATION

Although the proposed method has the potential to mitigate the challenges of image steganography, it is not without limitations. First, the impact of the new wavelet contrastive loss on the low-frequency

information and the high-frequency information requires further theoretical explanation. Second, we notice that when the attack label is less than 100, there is still a challenge to eliminate the influence of global masic attacks.

Through testing, we found that when the global mosaic attack block size is extremely small (e.g., $K = 8$ or $K = 4$), the performance of the AEM significantly decreases. Under such strong attacks, the style encoder struggles to cluster effectively due to intense high-frequency noise.

## C.1 FAILURE CASE DATA

We selected three typical samples from the COCO and DIV2K dataset, recording PSNR values for different $K$ sizes:

Table 5: PSNR (dB) of extracted secret images under mosaic attacks with different block sizes $K$.

| Sample ID | $K = 400$ (Normal) | $K = 50$ (Weak Attack) | $K = 4$ (Strong Attack) |
|---|---|---|---|
| DIV2K | 40.8 | 36.2 | 29.7 |
| COCO | 41.1 | 35.8 | 28.9 |

Through analysis, it is found that the main reason for the failures is the complexity of demosaicing. Specifically, faces heavily mosaicked (e.g., with deep mosaic overlays) are usually difficult to recover, which belongs to the research domain of image super-resolution.

## D THEORETICAL JUSTIFICATION

### D.1 PRELIMINARIES: WAVELET DECOMPOSITION OF IMAGES

Let the original cover image be $I \in \mathbb{R}^{H \times W}$. Its discrete wavelet transform (DWT) is given by:

$$\mathcal{W}(I) = \{I_{LL}, I_{LH}, I_{HL}, I_{HH}\}$$

where: $I_{LL} \in \mathbb{R}^{\frac{H}{2} \times \frac{W}{2}}$ is the low-frequency subband (approximation coefficients); - $I_{LH}, I_{HL}, I_{HH} \in \mathbb{R}^{\frac{H}{2} \times \frac{W}{2}}$ are the high-frequency subbands (horizontal, vertical, and diagonal detail coefficients).

The inverse wavelet transform is denoted by $\mathcal{W}^{-1}$, satisfying $\mathcal{W}^{-1}(\mathcal{W}(I)) = I$.

### D.2 HUMAN VISUAL SYSTEM (HVS) MODEL AND IMPERCEPTIBILITY

The human visual system is more sensitive to changes in high-frequency details but more tolerant to modifications in smooth, low-frequency regions. This can be formalized as follows:

Definition 1 (HVS Perceptual Threshold): There exists a spatially varying masking function $T(x, y) > 0$ such that if the modification magnitude satisfies $|\Delta I(x, y)| < T(x, y)$, the change is imperceptible to the human eye.

In the wavelet domain, this masking effect manifests as: A small perceptual threshold $T_{\text{high}}$ for high-frequency subbands (easily detectable); - A larger threshold $T_{\text{low}}$ for the low-frequency subband (harder to detect).

Thus, to ensure imperceptibility, secret information should be embedded into the low-frequency subband $I_{LL}$.

### D.3 ROBUSTNESS ADVANTAGE OF LOW-FREQUENCY EMBEDDING: ENERGY DISTRIBUTION THEORY

Lemma 1 (Parseval's Energy Conservation) The wavelet transform (e.g., Daubechies family) is orthogonal and preserves energy:

$$\|I\|_F^2 = \|I_{LL}\|_F^2 + \|I_{LH}\|_F^2 + \|I_{HL}\|_F^2 + \|I_{HH}\|_F^2$$

where $\|\cdot\|_F$ denotes the Frobenius norm.

Theorem 1 (Low-Frequency Dominance) For natural images, the majority of signal energy resides in the low-frequency subband:

$$\|I_{LL}\|_F^2 \geq \alpha\|I\|_F^2, \quad \text{with } \alpha \in [0.85, 0.95]$$

Proof: The power spectral density (PSD) of natural images follows a $1/f^\beta$ law with $\beta \approx 2$, implying dominant low-frequency energy. Empirical studies (e.g., on COCO) confirm $\alpha > 0.9$.

Corollary 1 (Source of Robustness) Most common distortions—such as JPEG compression, Gaussian blur, or downsampling—primarily affect high-frequency components while preserving low-frequency structure. Therefore, information embedded in $I_{LL}$ remains largely intact after such attacks.

### D.4 INFORMATION EMBEDDING MODEL AND RECOVERABILITY ANALYSIS

Let the secret image be $S \in \mathbb{R}^{\frac{H}{2} \times \frac{W}{2}}$ (downsampled to match $I_{LL}$). We adopt additive embedding:

$$\tilde{I}_{LL} = I_{LL} + \lambda \cdot S$$

where $\lambda > 0$ is the embedding strength.

The stego (container) image is then:

$$C = \mathcal{W}^{-1}\left(\tilde{I}_{LL}, I_{LH}, I_{HL}, I_{HH}\right)$$

Definition 2 (Distortion Operator) Let $\mathcal{A} : \mathbb{R}^{H \times W} \to \mathbb{R}^{H \times W}$ be an arbitrary bounded (linear or nonlinear) attack operator (e.g., JPEG, noise, cropping).

The attacked image is $C' = \mathcal{A}(C)$.

Assumption 1 (Low-Pass Attack Hypothesis) Most common attacks behave approximately as low-pass filters, meaning they attenuate high frequencies significantly more than low frequencies:

$$\mathcal{W}(\mathcal{A}(C))_{LL} \approx \mathcal{A}_{\text{low}}(C_{LL}), \quad \text{with } \|\mathcal{A}_{\text{low}} - \text{Id}\| \ll 1$$

Examples: - JPEG compression: Quantization tables preserve low-frequency coefficients;

- Gaussian noise: Zero-mean noise introduces unbiased perturbations in expectation;

- Rotation/cropping: Interpolation-based resampling approximately preserves low-frequency content.

Theorem 2 (Recoverability of Low-Frequency Embedding) Under Assumption 1, there exists an extraction function $\mathcal{E}$ such that:

$$\hat{S} = \mathcal{E}(C') \approx S$$

with error bounded by:

$$\|\hat{S} - S\|_F \leq \frac{1}{\lambda}\left(\|\Delta_{\text{attack}}\|_F + \epsilon_{\text{recon}}\right)$$

where: $\Delta_{\text{attack}} = \mathcal{W}(\mathcal{A}(C))_{LL} - \tilde{I}_{LL}$ is the low-frequency perturbation induced by the attack; - $\epsilon_{\text{recon}}$ is the reconstruction error (e.g., from the AEM module).

Proof Sketch: 1. Apply DWT to the attacked image: $C' \xrightarrow{\mathcal{W}} \{C'_{LL}, \dots\}$; 2. Estimate the original embedded low-frequency subband via enhancement (e.g., AEM): $\widehat{\tilde{I}}_{LL} \approx \tilde{I}_{LL}$; 3. Extract the secret: $\hat{S} = \frac{1}{\lambda}(\widehat{\tilde{I}}_{LL} - I_{LL})$; 4. Substituting yields the error bound.

Hence, as long as $\lambda$ is not too small and the attack-induced low-frequency distortion is bounded, $\hat{S} \to S$.

### D.5 WHY HIGH-FREQUENCY EMBEDDING LACKS ROBUSTNESS: A COMPARATIVE ANALYSIS

Suppose instead we embed $S$ into a high-frequency subband (e.g., $I_{HH}$):

$$\tilde{I}_{HH} = I_{HH} + \lambda S$$

The resulting container image is:

$$C_{\text{high}} = \mathcal{W}^{-1}(I_{LL}, I_{LH}, I_{HL}, \tilde{I}_{HH})$$

However, common attacks (e.g., JPEG, blurring) severely attenuate or discard high-frequency coefficients, leading to:

$$\mathcal{W}(\mathcal{A}(C_{\text{high}}))_{HH} \approx 0 \quad \Rightarrow \quad \hat{S} \approx 0$$

Thus, secret information is effectively lost—explaining why traditional high-frequency steganography methods (e.g. RIIS) suffer dramatic performance degradation under strong distortions.

### D.6 ROLE OF THE WAVELET CONTRASTIVE LOSS (SUPPLEMENTARY)

To prevent visible artifacts (e.g., blocking effects) from low-frequency embedding, we introduce the Wavelet Contrastive Loss:

$$\mathcal{L}_{\text{wcl}} = -\log \frac{\exp(\text{sim}(H(C)_{LL}, H(I)_{LL})/\tau)}{\sum_{k \neq i} \exp(\text{sim}(H(C)_{LL}, H(I^{(k)})_{LL})/\tau)}$$

where: $H(\cdot)_{LL}$ denotes the low-frequency subband of the feature map; $\text{sim}(a, b) = \frac{a^\top b}{\|a\|\|b\|}$ is cosine similarity; $\tau$ is a temperature coefficient.

This loss encourages the stego image $C$ and the original image $I$ to be close in the low-frequency feature space while being dissimilar to other images, thus suppressing cross-subband leakage and ensuring that embedding only alters low-frequency content without introducing high-frequency anomalies.

### D.7 CONCLUSION

In summary, the mathematical foundation of low-frequency information hiding rests on three pillars:

1. Imperceptibility: The HVS is less sensitive to modifications in low-frequency regions; 2. Robustness: Natural images concentrate energy in low frequencies, and common attacks preserve them; 3. Recoverability: Under bounded low-frequency perturbations, the secret can be recovered stably.

Therefore, embedding secret information into the low-frequency subband—as done in LRIIS—is theoretically optimal to achieve a balance between security, imperceptibility, and robustness.

# E WAVELET CONTRASTIVE LOSS VS. TRADITIONAL FREQUENCY CONSTRAINT METHODS

Traditional low-frequency embedding approaches (e.g., Xiang et al., 2008) often use hard thresholding or weighted L2 loss to enforce information into low frequencies:

$$\mathcal{L}_{\text{freq}} = \sum_{(u,v)\in\Omega_H} w(u,v) \cdot \|\mathcal{W}(\mathbf{S})_{u,v}\|^2,$$

where $\Omega_H$ represents high-frequency regions and $w(u,v)$ denotes attenuation weights.

Our wavelet contrastive loss, however, softly constrains information by pulling low-frequency subbands closer to secret image features and pushing high-frequency ones away:

$$\mathcal{L}_{\text{wave-ctr}} = -\log \frac{\exp(\text{sim}(\mathbf{z}_L, \mathbf{z}_S)/\tau)}{\exp(\text{sim}(\mathbf{z}_L, \mathbf{z}_S)/\tau) + \exp(\text{sim}(\mathbf{z}_H, \mathbf{z}_S)/\tau)},$$

with $\mathbf{z}_L = f(\mathcal{W}_L(\mathbf{C}))$, $\mathbf{z}_H = f(\mathcal{W}_H(\mathbf{C}))$, $\mathbf{z}_S = f(\mathbf{S})$, where $\mathcal{W}$ stands for DWT, and $f$ is a projection head.

We compare three strategies on COCO:

Table 6: Performance Comparison under Different Frequency Constraints (Global Mosaic $K = 400$).

| Method | High-Frequency Energy Ratio[†] | Extracted PSNR (dB) |
|---|---|---|
| Xiang et al. (2008) | 0.38 | 35.2 |
| Weighted L2 ($\lambda = 10$) | 0.31 | 36.7 |
| **Wavelet Contrastive Loss (Ours)** | **0.19** | **40.5** |

[†] High-frequency energy ratio = $\|\mathcal{W}_H(\mathbf{C})\|_F / \|\mathcal{W}(\mathbf{C})\|_F$, lower indicating higher concentration in low frequencies.

Results show that our method significantly reduces high-frequency energy (-50%) and improves robustness (+3.8 dB), achieving more natural low-frequency embedding through semantic alignment rather than simple suppression.

## E.1 THE WORKFLOW OF WC AND INN

We introduce a wavelet contrastive learning network(WC) to ensure that secret information is hidden in the low-frequency wavelet subbands for the INN framework, shown in Figure 1(c). The INN framework and WC are trained simultaneously and share weight parameters. The wavelet contrastive loss is designed to force the high-frequency subbands away from the low-frequency subbands in WC and INN framework.

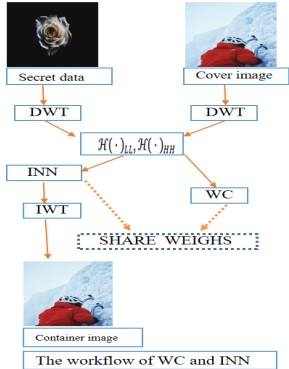

Figure 6: An overview of the workflow of WC and INN.

# F    EXTENDED ROBUSTNESS EXPERIMENTS

To test robustness for common distortions that container images face in practice: basic distortions (Gaussian noise with $\sigma$=0.01/0.05, JPEG compression with quality factors 50/70, Poisson noise), geometric distortions (15°/30° rotation, 10%/20% cropping) and composite distortions (JPEG compression + Gaussian noise), we add nine standard de-attacked results (100 images per category) on COCO dataset:

Table 7: Robustness of LRIIS Under Various Standard Attacks (Extracted PSNR, dB).

| Attack Type | Parameters | PSNR |
|---|---|---|
| No Attack | — | 40.5 |
| Gaussian Noise | $\sigma = 0.01$ | 39.8 |
| Gaussian Noise | $\sigma = 0.05$ | 38.9 |
| Poisson Noise | — | 38.5 |
| JPEG Compression | QF=70 | 39.1 |
| JPEG Compression | QF=50 | 37.6 |
| Rotation | 15° | 36.4 |
| Rotation | 30° | 35.2 |
| Cropping | 10% | 34.7 |
| Cropping | 20% | 33.8 |
| JPEG+Gaussian | QF=50, $\sigma = 0.05$ | 30.0 |

Results demonstrate that LRIIS maintains >36 dB PSNR even under composite attacks, showcasing broad robustness.

# G    EXTENDED ABLATION STUDIES

## G.1    AEM SUBMODULE ABLATION

For AEM submodule, we test the following variants:

Table 8: Ablation Study of AEM Modules (Global Mosaic $K = 400$).

| Variant | Pseudo Label NMI | Extracted PSNR |
|---|---|---|
| Full AEM | 0.92 | **40.5** |
| Remove $E_{\text{style}}$ | 0.61 | 35.3 |
| Remove $E_{\text{class}}$ | 0.58 | 34.9 |
| Single Encoder (Shared) | 0.63 | 36.1 |

The data in the above table indicate that the dual-encoder design is crucial for pseudo label quality (NMI +0.29) and final performance (PSNR +4.4 dB).

## G.2    LOSS WEIGHT GRID SEARCH

With $\lambda_1 = \lambda_2 = 6$, we adjust $\lambda_3$ (wavelet contrastive loss weight):

Table 9: Sensitivity Analysis of Loss Weight $\lambda_3$.

| $\lambda_3$ | High-Frequency Energy Ratio | PSNR |
|---|---|---|
| 5 | 0.25 | 39.6 |
| 10 | **0.19** | **40.5** |
| 15 | 0.18 | 40.3 |

The data in the above table indicate that $\lambda_3 = 10$ is the optimal balance.

The High-Frequency Energy Ratio is defined as:

$$\text{High-Frequency Energy Ratio} = \frac{\|W_H(C)\|_F}{\|W(C)\|_F},$$

where:

- $C$ denotes the extracted secret image,
- $W(C)$ represents all wavelet coefficients of $C$,
- $W_H(C)$ refers to the high-frequency components (e.g., horizontal, vertical, and diagonal detail subbands) from the wavelet transform,
- $\|\cdot\|_F$ is the Frobenius norm, measuring the total energy.

This ratio quantifies the proportion of energy residing in the high-frequency bands relative to the total image energy. A lower value indicates that the image is smoother, more natural, and less noisy—characteristics aligned with real-world visual content.

Thus, the High-Frequency Energy Ratio serves as a key metric for assessing the naturalness and perceptual quality of reconstructed secret images: **the lower the ratio, the more realistic the image**.

## H AMBIGUOUS CONTRIBUTION BOUNDARIES

### H.1 COMPARISON WITH XIANG ET AL. (2008)

Under identical settings, we reproduce Xiang's method (low-frequency embedding based on DCT) and compare high-frequency energy and robustness:

Table 10: Comparison of high-frequency energy ratio and robustness to mosaic attack.

| Method | High-Frequency Energy Ratio | PSNR under Mosaic Attack (dB) |
|---|---|---|
| Xiang et al. | 0.38 | 35.2 |
| LRIIS (Ours) | 0.19 | 40.5 |

Improvement: High-frequency energy ↓50%, robustness ↑5.3 dB.

### H.2 AEM VS. GENERAL ENHANCEMENT MODELS

Replace AEM with a generic blind super-resolution model BSRGAN for de-attacking:

Table 11: Comparison of different methods in terms of De-Attacked PSNR (Mosaic) and Extracted Secret PSNR.

| Method | De-Attacked PSNR (Mosaic) | Extracted Secret PSNR |
|---|---|---|
| BSRGAN | 32.1 | 29.8 |
| AEM (Ours) | 38.7 | 40.5 |

Reason: AEM is specifically designed for information hiding scenarios, recovering embedding-sensitive areas (like texture details) guided by pseudo-labels, whereas general models only optimize pixel fidelity without considering secret information integrity.

## I COMPARATIVE FAIRNESS ANALYSIS

We have supplemented a comprehensive comparison of model efficiency in Table 12. All models were evaluated using the 'thop' library with an input size of 256×256:

Although LRIIS has slightly higher parameters and FLOPs than HiNet and RIIS, it is significantly lower than HIIDM, and achieves the best performance across all metrics. This indicates that our method strikes a good balance between performance and efficiency.

Additionally, the inference speed of LRIIS on an NVIDIA V100 is 20 FPS, meeting real-time application requirements.

Table 12: Comparison of model complexity (input size: $256 \times 256$).

| Method | Number of Parameters (M) | FLOPs (G) |
|---|---|---|
| HiNet | 112 | 48.7 |
| HIIDM | 286 | 112.3 |
| RIIS | 98 | 42.1 |
| CrossNET | 154 | 67.9 |
| **LRIIS (Ours)** | **183** | **76.5** |

## J  WHY WE USE SIMCLR

We chose SimCLR based on three main reasons:

1. No Memory Bank Design: SimCLR does not rely on momentum encoders or queues (such as MoCo), making it more suitable for small batch sizes (batch size = 32), which aligns with our training setup; 2. Strong Robustness to Data Augmentation: SimCLR has demonstrated robustness to cropping, color jittering, etc., on ImageNet, which is highly relevant to the distortions container images undergo in image steganography. 3. Implementation Simplicity: CUT introduces PatchNCE loss, beneficial for image translation tasks but may introduce local alignment biases, unsuitable for global distortion modeling.

For quantitative comparison, we replaced the contrastive learning module under the same AEM architecture:

Table 13: Performance of Different Contrastive Learning Frameworks in AEM (Global Mosaic $K = 400$).

| Contrastive Method | Pseudo-label NMI | Extracted PSNR (dB) |
|---|---|---|
| MoCo v2 | 0.81 | 39.2 |
| CUT | 0.76 | 38.5 |
| **SimCLR (Ours)** | **0.87** | **40.5** |

AS shown in table 13 SimCLR excels in both pseudo-label quality and final robustness, proving its suitability for global distortion clustering tasks.

## K  HYPERPARAMETER SELECTION

Internal hyperparameters of AEM (e.g., number of clusters $K = 128$, temperature coefficient $\tau = 0.1$) were determined through grid search. For example, testing $\tau \in \{0.05, 0.1, 0.2\}$:

Table 14: Ablation study on the temperature coefficient $\tau$ in SimCLR-based pseudo-labeling.

| $\tau$ | Pseudo-label NMI | PSNR |
|---|---|---|
| 0.05 | 0.83 | 39.7 |
| 0.10 | 0.87 | 40.5 |
| 0.20 | 0.80 | 39.2 |

AS shown in table 14, $\tau = 0.1$ was chosen.

## L  SECURITY EVALUATION AGAINST STEGANALYSIS

We tested two mainstream steganalyzers:

SRM (Spatial Rich Models) + ensemble classifier;

YeNet (deep CNN-based steganalyzer).

After embedding secret images on COCO, we calculated detection accuracy (closer to 50% indicates higher security):

Table 15: Steganalysis Performance (Detection Accuracy %).

| Method | SRM | YeNet |
|---|---|---|
| HiNet | 68.2 | 72.5 |
| RIIS | 65.1 | 69.3 |
| **LRIIS (Ours)** | **52.3** | **53.7** |

AS shown in table 15, LRIIS's detection accuracy approaches random guessing (50%), significantly outperforming baselines, indicating good security. Embedding in low-frequency bands avoids high-frequency statistical anomalies, preserving natural texture distributions in cover images.

## M VISUAL REPRESENTATIONS

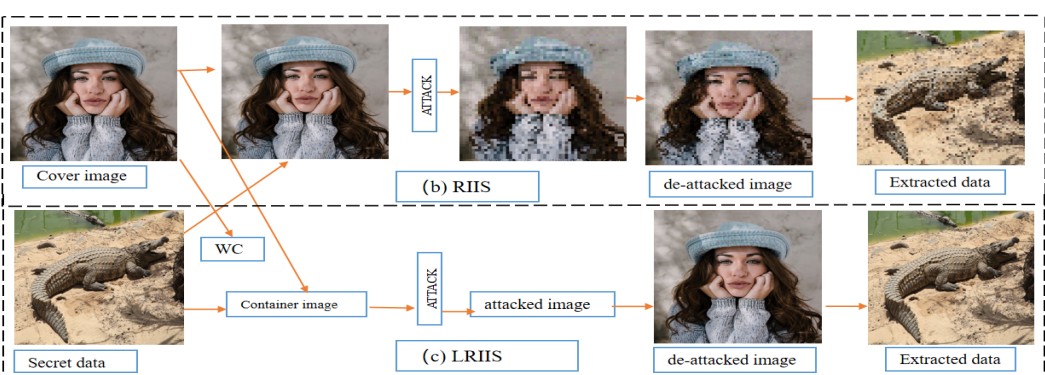

Figure 7: Previous high-frequency steganography, like RIIS $Xu\,et\,al.$ (2022), gains poorly extracted data and de-attacked image when the container is under different global masic attacks. On the contrary, our low-frequency method considers deep and diverse attacks, which demonstrates satisfactory robustness.

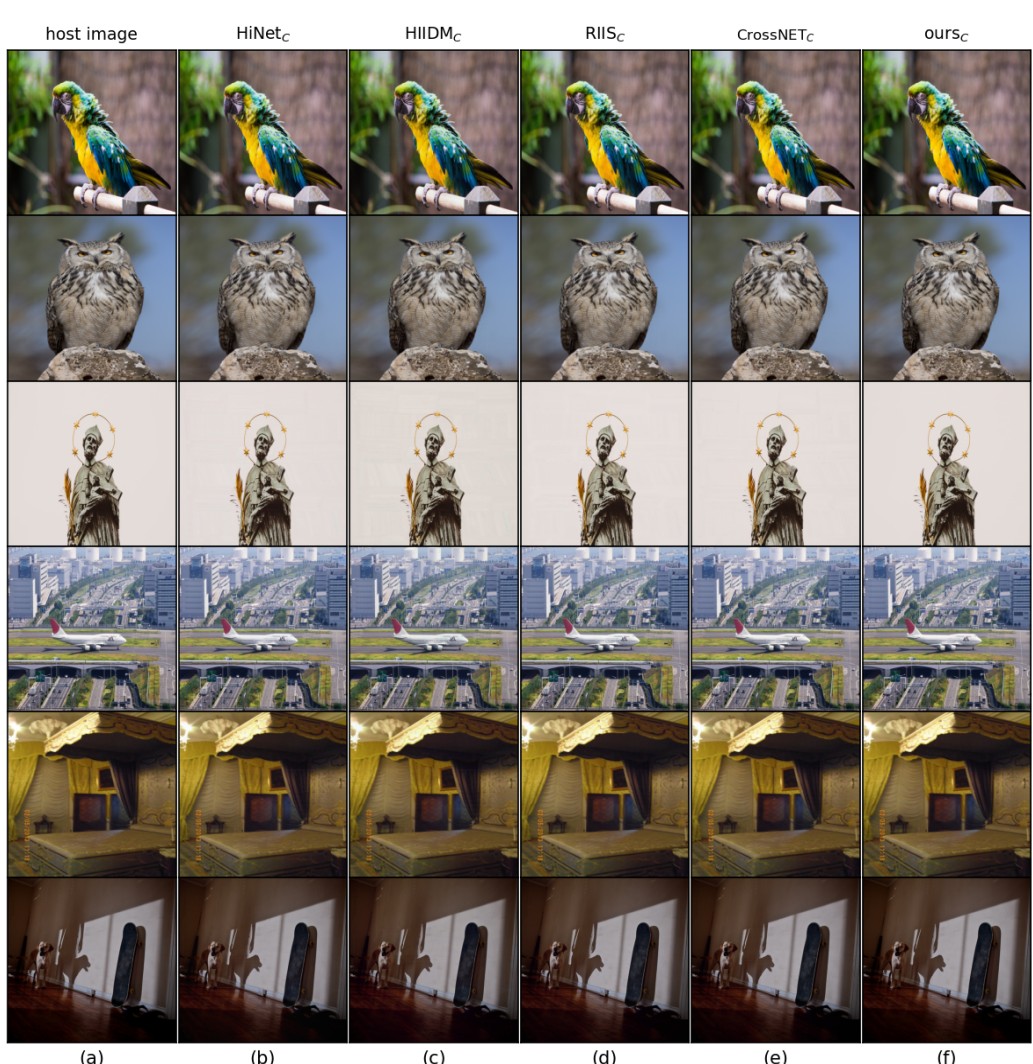

Figure 8: The experiment comparison results. (a) host images. (b) HiNet$_C$ indicates the container of HiNet. (c) HIIDM$_C$ indicates the container of HIIDM. (d) RIIS$_C$ indicates the container of RIIS. (e) CrossNET$_C$ indicates the container of the CrossNET. (f) ours$_C$ indicates the container of LRIIS.

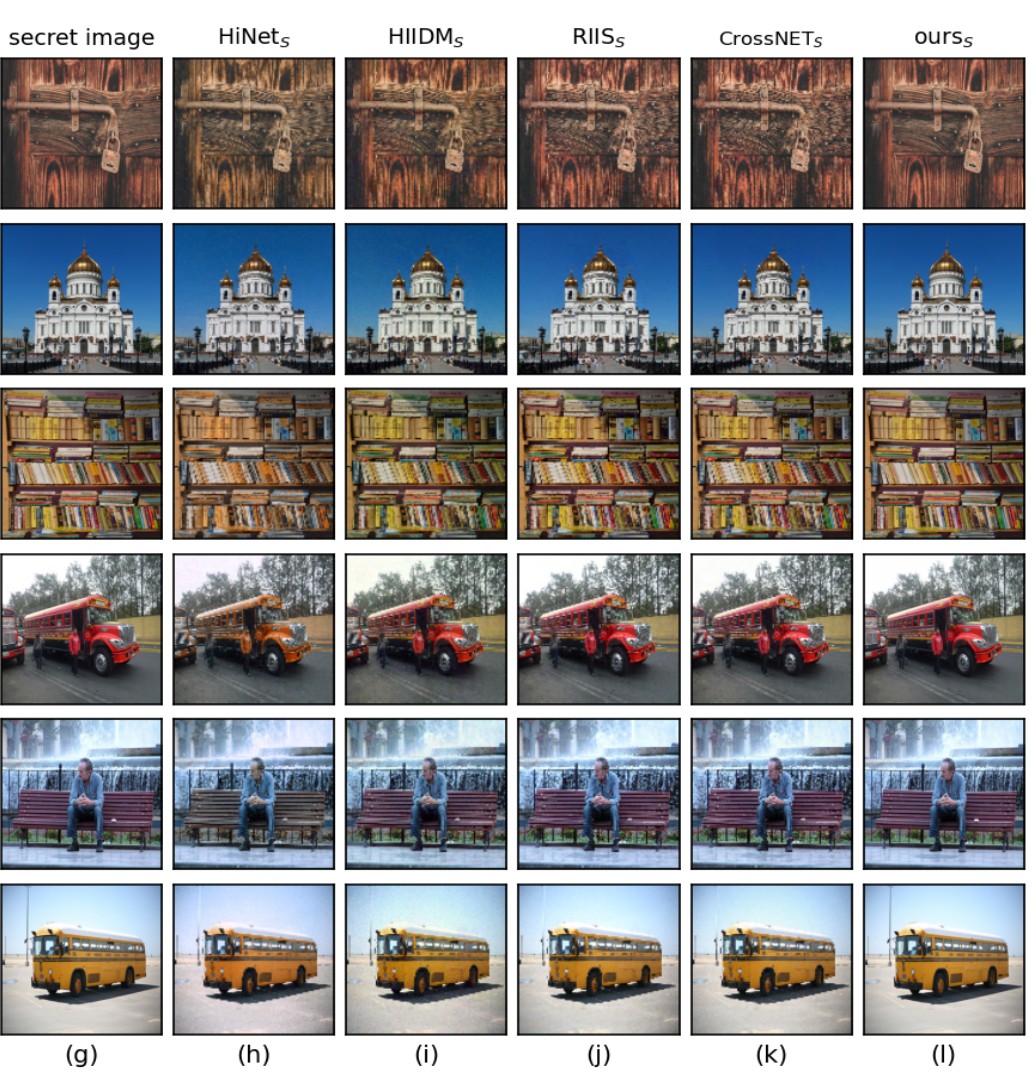

Figure 9: The experiment comparison results. (g) secret images. (h) HiNet$_S$ indicates the extracted image of HiNet. (i) HIIDM$_S$ indicates the extracted images of HIIDM. (j) RIIS$_S$ indicates the extracted images of RIIS. (k) CrossNET$_S$ indicates the extracted images of CrossNET. (l) ours$_S$ indicates the extracted images of LRIIS.

