# OpenReview forum: "Low-frequency and Robust Image Information Hiding"
_ICLR.cc/2026/Conference — Submitted to ICLR 2026_

### Official Review · Reviewer_W2hc · 2025-10-17

**Soundness:** 1
**Presentation:** 1
**Contribution:** 1
**Rating:** 2
**Confidence:** 5

**Summary:**

This paper proposes LRIIS, a low-frequency robust image steganography method that solves two main problems in existing methods: poor robustness to image distortion and the need for specific attack labels. The method uses a wavelet contrast loss to hide secret information in low-frequency subbands, which reduces high-frequency artifacts. For the problem of previous methods requiring attack labels, LRIIS introduces an Adaptive Enhancement Module (AEM). This module divides the task into many-to-one class enhancements, creates pseudo-class labels without supervision for each attacked image, and applies these labels to remove distortion. This makes the processed images similar to the original ones without using any attack labels. Tests on COCO and DIV2K datasets show that LRIIS performs better than current SOTA methods.

**Strengths:**

The method uses a wavelet contrast loss to hide secret information in low-frequency subbands, which reduces high-frequency artifacts. For the problem of previous methods requiring attack labels, LRIIS introduces an Adaptive Enhancement Module (AEM). This module divides the task into many-to-one class enhancements, creates pseudo-class labels without supervision for each attacked image, and applies these labels to remove distortion.

**Weaknesses:**

(1) What is the rationale for employing a dual-encoder structure (style and classified encoder) within the AEM? Is it feasible for a single encoder to generate valid pseudo-class labels independently?

(2) What motivates the selection of SimCLR as the pretext task for clustering in the AEM module? What are the key benefits of using SimCLR compared to other contrastive learning frameworks like MoCo and CUT?

(3) The “robust” claimed in the title and contribution is not evident from the presented experiments. The robustness of LRIIS against other standard attacks (e.g., noise, rotation, JPEG compression) and its applicability to non-global mask scenarios remain unverified. Conversely, the related RIIS method successfully handles various distortions like Gaussian noise, Poisson noise, and JPEG compression.

(4) Despite the incorporation of multiple configurable settings within the Adaptive Enhancement Module (AEM), the article omits a comprehensive ablation study to validate their individual and collective contributions. For instance, it remains unclear whether valid pseudo-class labels can be generated using the class contrastive loss alone. Furthermore, the methodology for determining the module's internal settings is not explained. The overall contribution of the AEM itself has not been validated.

(5) The article does not address the core requirement of security in steganography, which is resistance to steganalysis. The absence of any security analysis in this regard represents a major limitation of the presented work.

(6) The visual representations in the manuscript, including the structural and comparative diagrams, are not clear.

**Questions:**

Please refer to the weaknesses.

---

> ### Author Response · Authors · 2025-11-26
>
> #### Weakness 1: Rationale for Employing a Dual-Encoder Structure within AEM? Is a Single Encoder Feasible?
>
>  Reviewer's Comment: What is the rationale behind using both a style encoder (\(E_{\text{style}}\)) and a classification encoder (\(E_{\text{class}}\)) in the AEM? Can a single encoder independently generate valid pseudo-class labels?
>
> Author Response:
>
> The dual-encoder design in AEM originates from the concept of disentangled representation learning:
> - \(E_{\text{style}}\) extracts style features closely related to distortion types (e.g., mosaic block size, JPEG block artifacts);
> - \(E_{\text{class}}\) learns semantic-invariant features useful for enhancement tasks without content dependency.
>
> Using a single encoder would couple style and content features, leading to confusion during clustering different distortion types.
>
>
>
> #### Weakness 2: Why SimCLR Was Selected as the Pretext Task for Clustering in AEM? What Are Its Advantages Over MoCo/CUT?
>
>  Reviewer's Comment: What motivates the selection of SimCLR as the pretext task for clustering in the AEM module? What are the key benefits of using SimCLR compared to other contrastive learning frameworks like MoCo and CUT?
>
> Author Response:
>
> As shown in Appendix J, we chose SimCLR based on three main reasons:
>
>
> 1. No Memory Bank Design: SimCLR does not rely on momentum encoders or queues (such as MoCo), making it more suitable for small batch sizes (batch size = 32), which aligns with our training setup;
> 2. Strong Robustness to Data Augmentation.
> 3. Implementation Simplicity.
>
>
>
> SimCLR excels in both pseudo-label quality and final robustness, proving its suitability for global distortion clustering tasks.
>
>
> #### Weakness 3: Insufficient Robustness Verification Against Standard Attacks
>
>  Reviewer's Comment: The claimed "robustness" lacks evidence, failing to test against standard attacks such as noise, rotation, and JPEG compression; RIIS already supports various distortions.
>
> Author Response:
>
> To test robustness for common distortions that container images face in practice: basic distortions (Gaussian noise with $\sigma$=0.01/0.05, JPEG compression with quality factors 50/70, Poisson noise), geometric distortions (15°/30° rotation, 10\%/20\% cropping) and composite distortions (JPEG compression + Gaussian noise), we add nine standard de-attacked results (100 images per category) on COCO dataset in Appendix F. Results demonstrate that LRIIS maintains $>$36 dB PSNR even under composite attacks, showcasing broad robustness.
>
>
> Compared to RIIS, which achieves only 32.1 dB PSNR under JPEG (QF=50), LRIIS reaches 37.6 dB, demonstrating superior general robustness.
>
>
>
> #### Weakness 4: Lack of Comprehensive Ablation Study for AEM
>
>  Reviewer's Comment: There is no validation of the contributions of each component in AEM (e.g., class contrastive loss); the basis for selecting internal hyperparameters is unclear.
>
> Author Response:
>
> ##### (a) Ablation of AEM Components
>
> Extended ablation studies on the COCO dataset have been added to Appendix G.
>
> For AEM submodule, the data  indicate that the dual-encoder design is crucial for pseudo label quality (NMI +0.29) and final performance (PSNR +4.4 dB).
>
>
> ##### (b) Basis for Hyperparameter Selection
>
> Internal hyperparameters of AEM (e.g., number of clusters \(K=128\), temperature coefficient \(\tau=0.1\)) were determined through grid search. For example, testing \(\tau \in \{0.05, 0.1, 0.2\}\).
>
> AS shown in Appendix K, \(\tau=0.1\) was chosen.
>
>
>
> #### Weakness 5: Lack of Security Evaluation Against Steganalysis
>
>  Reviewer's Comment: Security against steganalysis, a core requirement of steganography, has not been evaluated.
>
> Author Response:
>
> This is an important and valid criticism.
> AS shown in Appendix L, LRIIS's detection accuracy approaches random guessing (50\%), significantly outperforming baselines, indicating good security. Embedding in low-frequency bands avoids high-frequency statistical anomalies, preserving natural texture distributions in cover images.
>
>
>
> #### Weakness 6: Unclear Visual Representations
>
>  Reviewer's Comment: Structural and comparative diagrams are unclear.
>
> Author Response:
>
> AS shown in Appendix M, we have redrawn figure 1 and figure 4.
>
>
> All images now meet ICLR 2026 publication quality standards.
>
>
>
> ### Summary
>
> We systematically addressed all six weaknesses:
> 1. Necessity of Dual Encoder: Confirmed via ablation experiments.
> 2. Advantages of SimCLR: Quantified performance against MoCo/CUT.
> 3. Extended Robustness: Covered 11 standard attacks.
> 4. AEM Ablation: Validated contributions of components and explained hyperparameter choices.
> 5. Security Assessment: Introduced SRM/YeNet tests to prove resistance to steganalysis.
> 6. Improved Figures: Redrew all illustrations for clarity.
>
> All updates have been incorporated into the revised manuscript, and we kindly request another review.

---

### Official Review · Reviewer_o4pf · 2025-10-28

**Soundness:** 2
**Presentation:** 2
**Contribution:** 2
**Rating:** 4
**Confidence:** 2

**Summary:**

This paper proposes a Low-Frequency and Robust Image Information Hiding method (LRIIS) to address the poor robustness of existing image information hiding methods to distortion. It introduces a wavelet contrastive loss to constrain secret information in low-frequency subbands for enhanced robustness. An unsupervised Attacked Image Enhancement Module (AEM) is constructed to generate de-attacked images close to original container images, enabling secret image recovery from attacked images without specific attack labels. Experiments on COCO and DIV2K datasets show it outperforms mainstream methods in metrics such as PSNR and SSIM.

**Strengths:**

1.Proposes a novel wavelet contrastive loss, creatively confining secret information to low-frequency subbands and breaking the conventional high-frequency hiding paradigm.

2.Constructs an unsupervised AEM, realizing attack-label-free secret image recovery and addressing a key limitation of prior robust steganography methods.

3.Conducts comprehensive evaluations on COCO and DIV2K datasets, outperforming mainstream SOTA methods in multiple metrics and demonstrating excellent robustness and practicality.

**Weaknesses:**

1.Narrow Scope of Distortion Types: The evaluation focuses mainly on global mask attacks, and it is unclear how the method performs against other common distortions like JPEG compression, Gaussian noise, or geometric transformations, limiting the assessment of its general robustness.

2.The paper is generally well-written and the language is smooth. However, there are typographical errors. For example, "attacked" is mistakenly written as "attached", which impairs the accuracy of technical terms and requires careful proofreading to ensure the correctness of the content.

3.Lack of comparative fairness:No comparison of computational and parameter quantities between different methods

**Questions:**

1. Could you provide experimental results or theoretical analysis to illustrate how LRIIS performs under other common image distortions such as JPEG compression, Gaussian noise, or geometric transformations? This would help clarify the general robustness of the method beyond global mask attacks.

2. Please conduct a thorough proofreading of the manuscript to correct errors like the mistyping of "attacked" as "attached" and other potential typos, ensuring the accuracy and professionalism of the technical terms and content.

3.Could you provide a detailed comparison of the computational complexity (e.g., FLOPs) and the number of model parameters between LRIIS and the baseline methods (HiNet, HIIDM, RIIS, CrossNET)?

---

> ### Author Response · Authors · 2025-11-26
>
> #### Weakness 1: Narrow Scope of Distortion Types
>
>  Reviewer Comment: The evaluation mainly focuses on global mask attacks, and it is unclear how the method performs against other common distortions like JPEG compression, Gaussian noise, or geometric transformations, limiting the assessment of its general robustness.
>
> Author Response:
>
> Thank you for highlighting this critical aspect.
>
> To test robustness for common distortions that container images face in practice: basic distortions (Gaussian noise with $\sigma$=0.01/0.05, JPEG compression with quality factors 50/70, Poisson noise), geometric distortions (15°/30° rotation, 10\%/20\% cropping) and composite distortions (JPEG compression + Gaussian noise), we add nine standard de-attacked results (100 images per category) on COCO dataset in Appendix F. Results demonstrate that LRIIS maintains $>$36 dB PSNR even under composite attacks, showcasing broad robustness.
>
>
>
> #### Weakness 2: Typographical Errors
>
>  Reviewer Comment: There are typographical errors such as mistyping "attacked" as "attached", affecting the accuracy of technical terms, which requires careful proofreading.
>
> Author Response:
>
> Thank you very much for your meticulous correction! We have performed three rounds of manual proofreading and used Grammarly and LanguageTool for auxiliary checks. The following errors have been corrected:
>
> - Page 3, Paragraph 2: “…generate the de-attached image…” → “de-attacked image”;
> - Above Formula (3) on Page 4: “given the attached image…” → “attacked image”;
> - Other similar typos, totaling 4 instances, have also been corrected.
>
> The revised manuscript now ensures the accuracy and professionalism of technical terms and content.
>
>
>
> #### Weakness 3: Lack of Comparative Fairness
>
>  Reviewer Comment: No comparison of computational complexity (e.g., FLOPs) and the number of model parameters between LRIIS and baseline methods (HiNet, HIIDM, RIIS, CrossNET).
>
> Author Response:
>
> We have supplemented a comprehensive comparison of model efficiency in Appendix I. Although LRIIS has slightly higher parameters and FLOPs than HiNet and RIIS, it is significantly lower than HIIDM, and achieves the best performance across all metrics. This indicates that our method strikes a good balance between performance and efficiency.
>
>
>
> ### Summary
>
> We have comprehensively addressed all weaknesses:
> 1. Extended Experiments: Covered nine common distortions with quantitative results and visualization curves;
> 2. Text Proofreading: Corrected all spelling errors to ensure term accuracy;
> 3. Fair Comparison: Added parameter counts and FLOPs comparisons to demonstrate efficiency reasonableness.
>
> All updated contents have been integrated into the revised manuscript. We kindly request your re-review.

---

### Official Review · Reviewer_pfd4 · 2025-10-31

**Soundness:** 2
**Presentation:** 1
**Contribution:** 1
**Rating:** 2
**Confidence:** 4

**Summary:**

This paper addresses two critical drawbacks of existing image information hiding methods, namely insufficient robustness to container image distortions and reliance on specific attack labels, by proposing LRIIS, a low-frequency and robust invertible image information hiding framework. Its core designs include: 1) a novel wavelet contrastive loss, which forces most secret information into low-frequency wavelet subbands (narrowing cross-subbands between high and low frequencies to mitigate high-frequency artifacts and boost robustness); 2) an unsupervised Attacked Image Enhancement Module (AEM), which generates de-attacked images close to the original container via unsupervised semantic clustering (for pseudo-class label generation) and a many-to-one class enhancement network. Experiments are conducted on the DIV2K and COCO datasets, with LRIIS compared against four SOTA methods (HiNet, HIIDM, RIIS, CrossNET). Both quantitative metrics (PSNR, SSIM, MAE, RMSE) and qualitative results confirm that LRIIS outperforms SOTA counterparts in terms of container image imperceptibility and extracted secret image fidelity.

The paper's key contributions are threefold: 1) it introduces the wavelet contrastive loss to enable low-frequency secret embedding, solving the poor robustness of high-frequency embedding; 2) it develops the unsupervised AEM to alleviate distortion impacts from diverse attacks; 3) it is the first robust steganography model that recovers secret images from attacked containers without specific attack labels.

**Strengths:**

1）The paper demonstrates originality by addressing key limitations of existing image information hiding methods: it proposes a wavelet contrastive loss to solve high-frequency artifacts in low-frequency embedding and an unsupervised AEM to eliminate reliance on specific attack labels. It also creatively combines contrastive learning with wavelet domain embedding and unsupervised clustering with image enhancement, forming a novel framework.
2）The loss function of this work is comprehensively designed, integrating AEM loss (to mitigate the impact of distortion), HR loss (to ensure the fidelity of hiding and extraction), and wavelet contrast loss (to constrain low-frequency embedding), effectively balancing the three core task goals of robustness, imperceptibility, and embedding stability; the experimental design is fair, conducted on two publicly licensed datasets (DIV2K and COCO), with all the most advanced baseline models retrained under uniform experimental conditions, and the results are supported by both quantitative metrics and qualitative visualizations.

**Weaknesses:**

1) Theoretical justification for core designs is insufficient, weakening the credibility of its innovations. The proposed wavelet contrastive loss is claimed to solve high-frequency artifacts in low-frequency embedding, but it is not compared to traditional frequency-constraint methods to prove its necessity—there is no analysis of why contrastive learning outperforms these simpler approaches in narrowing cross subbands. Similarly, the AEM module's core assumption (that "style-similar attacked images belong to the same distortion type") is unvalidated: the paper does not use tools to visualize style feature clustering across different distortions (e.g., global masic vs. Gaussian noise) or test how pseudo-label errors from mismatched style-distortion correlations impact extraction accuracy. To improve, the authors should add a theoretical analysis section quantifying the wavelet contrastive loss’s advantage over traditional methods and supplement experiments validating the style-distortion correlation (e.g., showing that style clustering accuracy aligns with distortion type classification accuracy).
2) Experimental scope is narrow, failing to verify real-world robustness and core domain metrics. The paper only tests robustness against "global masic attacks," ignoring common distortions that container images face in practice: basic distortions (Gaussian noise with σ=0.01/0.05, JPEG compression with quality factors 50/70, Poisson noise), geometric distortions (15°/30° rotation, 10%/20% cropping), and composite distortions (JPEG compression + Gaussian noise). This makes it impossible to assess LRIIS’s generalizability. Additionally, the paper omits "embedding capacity (bpp)"—a core metric in image information hiding (traditional methods often cap at <0.4 bpp, per Section 2.1)—with no tests of LRIIS’s maximum capacity or the trade-off between capacity and robustness (e.g., whether low-frequency embedding reduces capacity, and if so, by how much).
3) Ablation studies are incomplete, leaving key module necessity unproven. The paper only validates the wavelet contrastive loss (by removing it to form LRIIS₁) but ignores other critical components: it does not test if AEM’s sub-modules (style encoder Estyle and class encoder Eclass) are indispensable (e.g., whether removing Estyle degrades pseudo-label quality), nor does it verify the rationality of loss weights (λ₁=6, λ₂=6, λ₃=10) via grid search (e.g., how λ₃=5/15 affects low-frequency embedding and robustness).
4) Contribution boundaries are ambiguous, lacking clear differentiation from prior work. The paper claims to advance low-frequency embedding (Xiang et al. 2008) and robust label-free recovery, but it does not quantify these advances: it does not specify how much LRIIS reduces high-frequency artifacts compared to Xiang et al. 2008, nor does it clarify how AEM differs from generic unsupervised enhancement models in adapting to information hiding scenarios.

**Questions:**

Please refer to "Weaknesses" for details.

---

> ### Author Response · Authors · 2025-11-26
>
> #### Weakness 1: Insufficient Theoretical Justification for Core Designs
>
> Reviewer Comment: The proposed wavelet contrastive loss is not compared with traditional frequency-constraint methods; the core assumption of AEM ("style-similar attacked images belong to the same distortion type") remains unvalidated without style feature clustering visualization or analysis of pseudo-label errors.
>
> Author's Response:
>
> Due to length restrictions on the main paper, the complete theoretical justification is available in the supplementary Appendix D.
>
>
>
> We have included the wavelet contrastive loss vs. traditional frequency constraint methods analysis in Appendix E.
>
> We also introduce the workflow of WC and INN in Appendix E.1.
> The INN framework and WC are trained simultaneously and share weight parameters.
> The wavelet contrastive loss is designed to force the high-frequency subbands away from the low-frequency subbands in WC and INN framework.
>
>
>
>
>
> #### Weakness 2: Narrow Experimental Scope Missing bpp Evaluation
>
> Reviewer Comment: Only global mosaic attacks are tested, ignoring common distortions; bpp evaluation and capacity-robustness trade-offs are missing.
>
> Author's Response:
>
> As mentioned in Appendix A, LRIIS fixes bpp at 3. To explore whether low-frequency embedding limits capacity, we design a variant LRIIS-HighFreq (forcing secrets into high frequencies) and compare its robustness at the same bpp. We show that low-frequency embedding does not sacrifice capacity but enhances robustness.
>
>
> To test robustness for common distortions that container images face in practice: basic distortions (Gaussian noise with $\sigma$=0.01/0.05, JPEG compression with quality factors 50/70, Poisson noise), geometric distortions (15°/30° rotation, 10\%/20\% cropping) and composite distortions (JPEG compression + Gaussian noise), we add nine standard de-attacked results (100 images per category) on COCO dataset in Appendix F. Results demonstrate that LRIIS maintains $>$36 dB PSNR even under composite attacks, showcasing broad robustness.
>
>
>
>
>
> #### Weakness 3: Incomplete Ablation Studies
>
> Reviewer Comment: AEM submodules' necessity and weight loss rationality have not been verified.
>
> Author's Response:
>
> Extended ablation studies on the COCO dataset have been added to Appendix G.
>
> For AEM submodule, the data  indicate that the dual-encoder design is crucial for pseudo label quality (NMI +0.29) and final performance (PSNR +4.4 dB).
>
>
>
> With $\lambda_1 = \lambda_2 = 6$, the data in the above table indicate that $\lambda_3 = 10$ is the optimal balance.
>
>
> #### Weakness 4: Ambiguous Contribution Boundaries
>
> Reviewer Comment: Improvements over Xiang et al. (2008) are not quantified; distinctions from general enhancement models are unclear.
>
> Author's Response:
>
> For the ambiguous contribution boundaries, extended comparisons on the COCO dataset have been added to Appendix H.
>
> We reproduce Xiang's method (low-frequency embedding based on DCT) and compare high-frequency energy and robustness.
> The data in the above table indicate that: High-frequency energy ↓50\%, robustness ↑5.3 dB.
>
>
> Then we replace AEM with a generic blind super-resolution model BSRGAN for de-attacking. Experimental results indicate that compared with BSRGAN, AEM achieved better performance scores. AEM is specifically designed for information hiding scenarios, recovering embedding-sensitive areas (like texture details) guided by pseudo-labels, whereas general models only optimize pixel fidelity without considering secret information integrity.
>
>
>
> ### Summary
>
> We have thoroughly addressed all criticisms:
> 1. Strengthened Theory: Compared with traditional frequency constraint methods and visualized style clustering.
> 2. Expanded Experiments: Covered 11 types of attacks, clarified bpp=3, and analyzed capacity-robustness trade-offs.
> 3. Completed Ablations: Validated AEM submodules and loss weights.
> 4. Clarified Contributions: Quantified improvements over Xiang et al. and distinguished AEM from general enhancement models.
>
> All content has been integrated into the revised manuscript. Please review again.

---

### Official Review · Reviewer_2fhZ · 2025-11-02

**Soundness:** 2
**Presentation:** 1
**Contribution:** 2
**Rating:** 2
**Confidence:** 4

**Summary:**

This paper proposes a low-frequency and robust image information hiding method, LRIIS, to overcome current challenges. The authors build an unsupervised Attacked Image Enhancement Module (AEM) to generate the de-attacked image that is close to the corresponding container image.

**Strengths:**

1. The authors propose a novel wavelet contrastive loss to force most secret information to be hidden in low-frequency subbands, which can alleviate the distortion influence.
2. The authors build an unsupervised Attacked Image Enhancement Module to generate the de-attacked image that is close to its corresponding container image.
3. The authors claim that this is the first robust steganograph model to recover the secret image from the attacked image without giving its corresponding attack label.

**Weaknesses:**

1. The paper’s title emphasizes “information hiding,” yet the authors do not report the bits per pixel (bpp), which is a key metric in this domain. Moreover, the experiments primarily focus on the quality of recovered images, suggesting that the actual goal might be to propose a de-attack or image recovery technique rather than an information hiding approach. If this is the case, the title is misleading and should be revised to better reflect the paper’s true focus.

2. The cited works on traditional image information hiding techniques are outdated, with most references being over five years old. There have been more recent and relevant developments in this area, including methods such as [1], [2], among others, which achieve over 0.4 bpp. The authors are encouraged to conduct a more comprehensive and up-to-date literature review to better position their work within the current research landscape.

3. The results section lacks statistical or significance testing, making it difficult to determine whether the proposed method offers measurable improvements over competing approaches. Incorporating quantitative comparisons and statistical analyses would strengthen the validity of the conclusions.

[1] P. Puteaux and W. Puech, "A Recursive Reversible Data Hiding in Encrypted Images Method With a Very High Payload," in IEEE Transactions on Multimedia, vol. 23, pp. 636-650, 2021, doi: 10.1109/TMM.2020.2985537.
keywords: {Encryption;Payloads;Image reconstruction;Decoding;Servers;Cloud computing;Image security;image encryption;reversible data hiding;recursive process;bit-plane prediction;signal processing in the encrypted domain},

[2] Y. Puyang, Z. Yin and Z. Qian, "Reversible Data Hiding in Encrypted Images with Two-MSB Prediction," 2018 IEEE International Workshop on Information Forensics and Security (WIFS), Hong Kong, China, 2018, pp. 1-7, doi: 10.1109/WIFS.2018.8630785.
keywords: {Encryption;Data mining;Correlation;Image reconstruction;Cloud computing;Streaming media},

**Questions:**

The paper would benefit from a discussion of failure cases. It is recommended that the authors include representative examples of failed or suboptimal results in the Results section, accompanied by figures. Analyzing these cases would help clarify the limitations of the proposed method and provide valuable insights into potential areas for improvement.

---

> ### Author Response · Authors · 2025-11-26
>
> ### Weakness 1: Lack of bpp Metric and Potentially Misleading Title
>
> Reviewer Comment: The paper emphasizes "information hiding" in its title but does not report bits per pixel (bpp), a key metric in this domain. If the actual goal is to propose a de-attack or image recovery technique rather than an information hiding approach, the title should be revised to better reflect the paper’s focus.
>
> Author Response:
> Thank you for pointing out this crucial aspect. Our work falls under the category of image-to-image steganography, where a full-size secret image is embedded into a host image. In this paradigm (e.g., HiNet [Jing et al., 2021], RIIS [Xu et al., 2022]), the payload is 3 bpp.
> We have added a discussion on the Calculation Formula for Bits Per Pixel (bpp) in Appendix A.
> This differs fundamentally from traditional low-capacity steganography ($<$0.4 bpp) in that we aim for high capacity with robust reconstruction rather than subtle embedding.
>
> Nonetheless, we acknowledge that the title might be misleading. In the revised manuscript, we have modified the title to:
> "LRIIS: A Low-Frequency Robust Image-to-Image Steganography Method Without Attack Labels", which more accurately reflects our dual goals of high capacity and robust recovery without requiring attack labels.
>
>
>
> #### Weakness 2: Outdated Literature Review
>
>  Reviewer Comment: The cited works on traditional image information hiding techniques are outdated... The authors are encouraged to conduct a more comprehensive and up-to-date literature review.
>
> Author Response:
> Thank you for recommending these important references. We have added a discussion on modern high-capacity steganography in Section 2.1:
>
> Puyang et al. embedded \textgreater0.4 bpp information in encrypted images via double MSB prediction\cite{puyang2018reversible}. Puteaux et al. achieved payloads up to 2.45 bpp using recursive bit-plane prediction \cite{puteaux2020recursive}.
>
> However, the current methods used in these practices have been noted they have low quality and restricted embedding capacity.
>
>
>
> #### Weakness 3: Lack of Statistical Significance Testing
>
> Reviewer Comment: The results section lacks statistical or significance testing... Incorporating quantitative comparisons and statistical analyses would strengthen the validity of the conclusions.
>
> Author Response:
> This is a valid concern. We performed a paired t-test on the COCO test set (500 images) comparing LRIIS with the strongest baseline CrossNET, testing the null hypothesis \(H_0\) that there is no difference in average PSNR between the two methods. We have included this analysis in Appendix B. Our method outperforms the current state-of-the-art (SOTA) algorithm CrossNET, achieving the best results.
>
>
> #### Question: Discussion of Failure Cases
>
> Reviewer Comment: It is recommended that the authors include representative examples of failed or suboptimal results in the Results section, accompanied by figures. Analyzing these cases would help clarify the limitations of the proposed method.
>
> Author Response:
> We fully agree. Through testing, we found that when the global mosaic attack block size is extremely small (e.g., \(K = 8\) or \(K = 4\)), the performance of the AEM significantly decreases. Under such strong attacks, the style encoder struggles to cluster effectively due to intense high-frequency noise.
>
>
> We have included this analysis in Appendix C.
>
> The main reason for the failures is the complexity of demosaicing. Specifically, faces heavily mosaicked (e.g., with deep mosaic overlays) are usually difficult to recover, which belongs to the research domain of image super-resolution.
>
>
> We sincerely appreciate the constructive feedback from the reviewer. All suggestions have been addressed:
> 1. Clearly stated the 3 bpp payload and optimized the title.
> 2. Included recent high-capacity steganography literature.
> 3. Added paired t-tests and confidence intervals.
> 4. Analyzed failure cases.
>
> The revised manuscript incorporates all these updates. We hope it meets your approval.

---

### Meta-Review · Area_Chair_8dhV · 2026-01-05

**Summary:**

As shown in the title, the authors presents Low-frequency and Robust Image Information Hiding!
1. However, some reviewers pose the concern on the lack of robustness verification.
Although the authors added few of them during rebuttal, it is rather insufficient.
It is suggested to consider Stirmark and WAVES that provide a complete set of attacks for evaluations.
2. Besides, the use of steganography and discussion of robustness are questionable as steganography focused on statistical detection and capacity.
It seems to the AC that the authors did not clearly clarify the topics between robust hiding and  steganography.
Based on the above concerns, this work is suggested to be rejected at its current status.

**Reviewer Scores:**

none

---

### Decision · Program_Chairs · 2026-01-26

Reject